# HOW TO WEIGHT MULTITASK FINETUNING? FAST PREVIEWS VIA BAYESIAN MODEL-MERGING

## ABSTRACT

When finetuning multiple tasks altogether, it is important to carefully weigh them to get a good performance, but searching for good weights can be difficult and costly. Here, we propose to aid the search with fast previews to quickly get a rough idea of different reweighting options. We use model merging to create previews by simply reusing and averaging parameters of models trained on each task separately (no retraining required). To improve the quality of previews, we propose a Bayesian approach to design new merging strategies by using more flexible posteriors. We validate our findings on vision and natural-language transformers. Our work shows the benefits of model merging via Bayes to improve multitask finetuning.

## 1 INTRODUCTION

Multitask finetuning has recently gained popularity due to the success of large pretrained models, but a careful weighting of tasks is crucial to get good performances (Liu et al., 2023; Xu et al., 2024; Chung et al., 2024). Similarly to the traditional multi-task learning (Caruana, 1997; Ruder, 2017), the weighting is useful in tackling data imbalance, task interference, negative transfer, and also effects of variable task difficulty (Raffel et al., 2020; Liu et al., 2023). When left unresolved, these can lead to issues, for instance, regarding safety (Jan et al., 2024). Weighting is also useful for *pre-finetuning* recently used for multi-lingual transfer and continual pretraining (Aghajanyan et al., 2021; Gemma Team, 2024a; Martins et al., 2024; Fujii et al., 2024).

Despite its importance, little has been done to address task weighting for multitask finetuning. For large models, an exhaustive search over weights is out of the question, but even if we could try a few weighting configurations, which ones should we try? There is no guide for that. Weights are often chosen arbitrarily and sometimes heuristically but these are not sufficient; see, for example, Liu et al. (2023, Sec. 6). The weighting methods used for deep learning and pretraining can be adapted to search for good weights (Ren et al., 2018; Chen et al., 2018; Raffel et al., 2020; Groenendijk et al., 2021; Du et al., 2022; Yan et al., 2022; Xie et al., 2023; Thakkar et al., 2023), but a quick guidance on reasonable search areas will still be useful to assist the search.

In this paper, we propose to aid the search with fast *previews* of performances, estimated to obtain weights that improve accuracy of multitask finetuning. We use model merging to create the previews where we train and store models on each task separately and reuse them later to create previews by simply averaging the model parameter for a wide-range of weights (Fig. 1). Our main contribution is a Bayesian approach to design new merging strategies that yield better previews over a wider range of weights. This differs fundamentally from previous work which only focus on the best performing weights (Don-Yehiya et al., 2023; Jiang et al., 2023; Feng et al., 2024; Stoica et al., 2024; Yang et al., 2024). Such strategies do not always yield good-quality previews; see Fig. 2a for an illustration.

We propose a Bayesian approach where more flexible posteriors yield better previews but with slightly higher costs (Figs. 2b and 2c). For instance, we show that Task Arithmetic (Ilharco et al., 2023) for LLMs corresponds to isotropic Gaussian posteriors, while better (and slightly more costly) Hessian-based methods (Daheim et al., 2024) employ more flexible Gaussian forms. We generalize this result to generic exponential-family posteriors and present a recipe to derive new merging strategies. We validate our method on several benchmarks on vision and natural-language transformers. For example, we experiment with image classification using Vision Transformers (ViTs) (Dosovitskiy et al., 2021) of 86M parameters and adding new languages to GEMMA with 2B parameters (Gemma Team, 2024b) for machine translation. Our results consistently show that more flexible posteriors

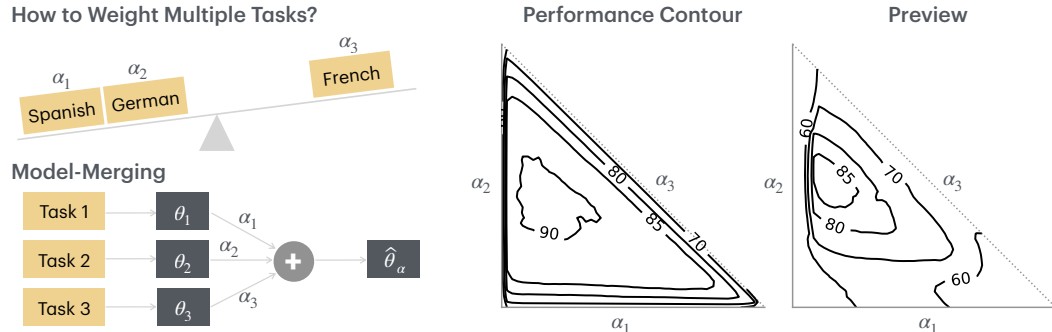

Figure 1: Our goal is to aid the search for good weights in weighted multitask finetuning. We show a performance contour for 3 tasks with weights $\alpha_1, \alpha_2$ and $\alpha_3$. The well performing regions are in the middle achieving around $90\%$ accuracy. We create a cheap preview of the contours by using model merging where previously trained models are quickly weighted with many $\boldsymbol{\alpha}$ values. The preview captures the rough shape of the true contours, encouraging a focus on the good regions.

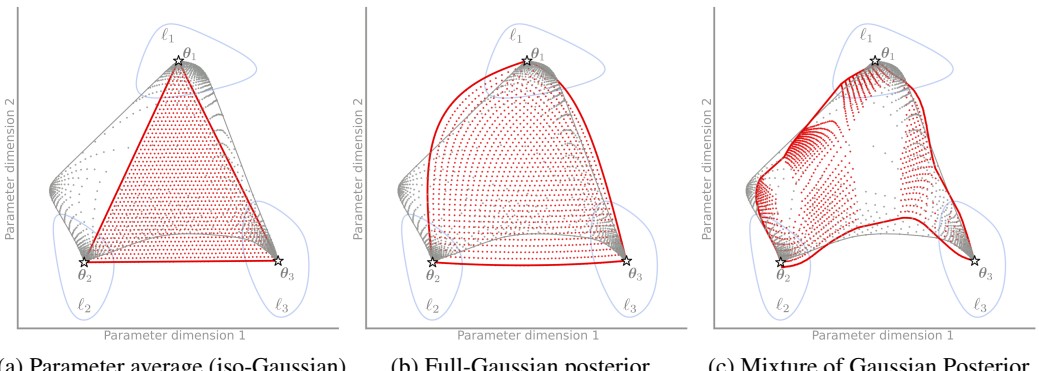

(a) Parameter average (iso-Gaussian)  (b) Full-Gaussian posterior  (c) Mixture of Gaussian Posterior

Figure 2: An illustration of our Bayesian approach to improve preview quality for a toy multitask-learning problem with 3 tasks. The losses $\ell_t$ defined over a 2-D $\boldsymbol{\theta}$ space and are weighted by $\alpha_t$ varied in a fixed grid over $[0, 1]$. Panel (a) shows that parameter averaging $\sum_t \alpha_t \boldsymbol{\theta}_t$ gives poor preview (red region) of the true performances (gray contour). Each dot corresponds to a weighting option. The quality is improved in panel (b) and (c) where merging strategies using full more flexible posteriors are used, respectively. The cost is slightly increased because they need Hessians and ensembles.

produce better previews and helps us choose weights to perform more accurate multitask finetuning. Our work combines ideas from model merging and Bayesian learning to improve multitask finetuning.

## 2 WEIGHTED MULTITASK FINETUNING

Multitask finetuning aims to finetune on multiple tasks altogether. For example, given a Large Language Model (LLM) trained for English, we may want to finetune it on multiple languages (Muennighoff et al., 2023), for instance, German, French, Chinese, Japanese, etc. Denoting the loss of each task by $\ell_t(\boldsymbol{\theta})$ for the model parameters $\boldsymbol{\theta}$, we want to finetune over a weighted loss

$$\sum_{t=1}^{T} \alpha_t \ell_t(\boldsymbol{\theta}), \text{ where } \alpha_t > 0 \text{ for } t = 1, 2, \ldots, T. \tag{1}$$

We denote the loss-weight vector by $\boldsymbol{\alpha} = (\alpha_1, \alpha_2, \ldots, \alpha_T)$ and the finetuned parameters obtained with the weighting by $\boldsymbol{\theta}_{\boldsymbol{\alpha}}$. We will generally assume that $\alpha_t$ sum to 1 but this is not strictly required. Such multitask finetuning has recently become important for LLM alignment and usability. For example, it is used for improving instruction-following abilities (Chung et al., 2024; Ouyang et al., 2022), different kinds of safety tuning (Gemma Team, 2024a), combining coding tasks (Liu et al.,

2023), and mixing coding and math skills into LLMs, which is useful even when they are designed for other tasks like machine translation (Martins et al., 2024).

In practice, it is important to choose $\boldsymbol{\alpha}$ carefully for reasons that are true for any multi-task learning problem (Caruana, 1997; Ruder, 2017). For instance, one issue is due to data-imbalance: different tasks may contain different types of information and some of higher quality than others. There is also task interference, for example, a model that does math well, may not necessarily be the best at languages. Additionally, learning some tasks might hurt the performances in the other tasks, and then there is variability in task difficulty: some tasks are harder to learn and we do not want those to impact the tasks that are relatively easier to learn.

The effects of these issues are often felt in practice. For example, adding too much safety data can make the model more conservative and reduce its usefulness (Bianchi et al., 2024); too much instruction finetuning can undo safety alignment and open new vulnerabilities (Qi et al., 2024; Jan et al., 2024). Such problems can be avoided by careful task weighting. Weighting is also useful during *pre-finetuning* where we try to balance multiple tasks differently during the last, say, $10\%$ of pretraining (Aghajanyan et al., 2021; Gemma Team, 2024a; Martins et al., 2024; Fujii et al., 2024).

Despite its importance, not much work has been done to find good ways to set the weights. Decades of work exist for multitask learning but multitask finetuning is a relatively new area and is still under-explored. Similarly to multi-task learning, an exhaustive search over the whole $\boldsymbol{\alpha}$-space is not feasible when $T$ is large. With little guidance, arbitrary values are tried to get an idea, for example, Fujii et al. (2024) try only two values of $\boldsymbol{\alpha}$ for continual pretraining. Sometimes heuristics are used and meta-learning approaches are also adopted, but the results are not always satisfactory, for example, see Liu et al. (2023, Sec. 6) who report such a result for LLMs trained for code generation.

Our goal in this paper is to provide a fast (and cheap) method to assist the search of good $\boldsymbol{\alpha}$ values. Such a guiding tool is useful to restrict the search to a few values and also to warm-start the optimization process. For this, we need a fast but accurate approach to quickly estimate the performance of $\boldsymbol{\theta}_{\boldsymbol{\alpha}}$ for a wide-range of $\boldsymbol{\alpha}$ values. Model merging is a useful tool for this, but the choice of merging strategy matters to get a high quality preview. We show that simple merging methods are not satisfactory because they can be quite inaccurate and may only yield good estimates for a small region in the $\boldsymbol{\alpha}$ space (see Fig. 2a). Our main technical contribution is to address this with a Bayesian approach to expand the region by designing a more accurate merging strategy. The quality is improved at the expense of cost but the approach still remains fast enough to be employed in practice. Existing methods on model merging in this space only focus on the best performing weights (Don-Yehiya et al., 2023; Jiang et al., 2023; Feng et al., 2024; Stoica et al., 2024; Yang et al., 2024). Our work instead aims to design merging strategies that work for a wide range of $\boldsymbol{\alpha}$ values.

## 3 FAST PREVIEWS VIA BAYESIAN MODEL-MERGING

Our goal is to create fast previews, that is, we want to estimate $\boldsymbol{\theta}_{\boldsymbol{\alpha}}$ obtained by finetuning over $\sum_t \alpha_t \ell_t$ for a wide-variety of $\boldsymbol{\alpha}$ values. The previews are useful to choose the $\boldsymbol{\alpha}$ that give the most accurate results for joint multitask finetuning over all tasks. Our approach does this in three steps:

1. Finetune $T$ models (denoted by $\boldsymbol{\theta}_t$) each separately over their own task $\ell_t(\boldsymbol{\theta})$.
2. Use Bayesian learning to build surrogate $\ell_t \approx \widehat{\ell}_t$ by using $\boldsymbol{\theta}_t$.
3. Create previews by finetuning over $\sum_t \alpha_t \widehat{\ell}_t$ for many $\boldsymbol{\alpha}$ values.

In step 2, we design accurate $\widehat{\ell}_t$ by using exponential-family posteriors. Such posterior always has a closed-form merging formula which enables step 3. We start by describing model merging.

### 3.1 MODEL MERGING AS A WEIGHTED-SURROGATE MINIMIZATION

Model merging uses simple parameter averaging but it can also be seen as minimization of a weighted sum of surrogates. For example, consider the simple averaging (SA) (Wortsman et al., 2022),

$$\widehat{\boldsymbol{\theta}}_{\boldsymbol{\alpha}}^{\text{SA}} = \sum_{t=1}^{T} \alpha_t \boldsymbol{\theta}_t \quad = \arg\min_{\boldsymbol{\theta}} \; \gamma \underbrace{\tfrac{1}{2}\|\boldsymbol{\theta}\|^2}_{\widehat{\mathcal{R}}_0(\boldsymbol{\theta})} + \sum_{t=1}^{T} \alpha_t \underbrace{\left(\ell_t(\boldsymbol{\theta}_t) + \tfrac{1}{2}\|\boldsymbol{\theta} - \boldsymbol{\theta}_t\|^2\right)}_{=\widehat{\ell}_t(\boldsymbol{\theta})}, \tag{2}$$

where the surrogate $\widehat{\ell}_t$ is a quadratic function. The term $\ell_t(\boldsymbol{\theta}_t)$ is a constant and can be ignored. The $\mathcal{R}_0$ is a regularizer with $\gamma = 1 - \sum_t \alpha_t$, which disappears if the $\alpha_t$ sum to 1. The equality can be verified by setting the derivative to zero. Other merging techniques can also be interpreted this way. For example, Task Arithmetic (TA) (Ilharco et al., 2023) finetunes over an LLM with parameters $\boldsymbol{\theta}_{\mathrm{LLM}}$ and can be seen as the following weighted-surrogate minimization with a regularizer,

$$\widehat{\boldsymbol{\theta}}_{\boldsymbol{\alpha}}^{\mathrm{TA}} = \boldsymbol{\theta}_{\mathrm{LLM}} + \sum_{t=1}^{T} \alpha_t(\boldsymbol{\theta}_t - \boldsymbol{\theta}_{\mathrm{LLM}}) \;\; = \arg\min_{\boldsymbol{\theta}} \; \gamma \tfrac{1}{2}\|\boldsymbol{\theta} - \boldsymbol{\theta}_{\mathrm{LLM}}\|^2 + \sum_{t=1}^{T} \alpha_t \tfrac{1}{2}\|\boldsymbol{\theta} - \boldsymbol{\theta}_t\|^2. \quad (3)$$

Here, we removed the constant $\ell_t(\boldsymbol{\theta}_t)$ for clarity. In general, many model merging methods can be interpreted as weighted-surrogate minimization, including Wortsman et al. (2022); Matena & Raffel (2022); Jin et al. (2023); Ortiz-Jimenez et al. (2023); Daheim et al. (2024).

The interpretation highlights a major source of error when estimating performance over a wide range of $\boldsymbol{\alpha}$ values: the accuracy of the surrogates. The surrogates used above can be seen as a simplistic Taylor approximation where we assume $\nabla \ell_t(\boldsymbol{\theta}_t)$ to be zero (due to local optimality) and Hessian $\nabla^2 \ell_t(\boldsymbol{\theta}_t)$ is set to identity,

$$\ell_t(\boldsymbol{\theta}) \approx \ell_t(\boldsymbol{\theta}_t) + \nabla \ell_t(\boldsymbol{\theta}_t)^\top (\boldsymbol{\theta} - \boldsymbol{\theta}_t) + \tfrac{1}{2}(\boldsymbol{\theta} - \boldsymbol{\theta}_t)^\top \nabla^2 \ell_t(\boldsymbol{\theta}_t)(\boldsymbol{\theta} - \boldsymbol{\theta}_t)$$
$$\approx \ell_t(\boldsymbol{\theta}_t) + \tfrac{1}{2}\|\boldsymbol{\theta} - \boldsymbol{\theta}_t\|^2. \quad (4)$$

The surrogates $\widehat{\ell}_t$ are tight at only one point and their inaccuracy increases as we move away from it. Model merging can be seen as using $\sum_t \alpha_t \widehat{\ell}_t$ (along with the regularizer $\widehat{\mathcal{R}}_0$) as a proxy to estimate the results of finetuning the original $\sum_t \alpha_t \ell_t$. However, when using a wide range of $\boldsymbol{\alpha}$ values, these inaccuracy can lead to poor estimates in some regions in the $\boldsymbol{\alpha}$ space. Essentially, the errors in different $\boldsymbol{\theta}$-regions become relevant and ultimately lead to a poor estimate.

Ideally, we would like the surrogates to be designed such that they are not only locally accurate but also in a wider region. We expect such surrogates to be globally accurate and beyond the point used in their definition, but how can be design them? Is there a general recipe to do so? We will now propose a Bayesian approach to answer these questions.

### 3.2 A BAYESIAN APPROACH TO MODEL MERGING

The surrogate minimization approach can be seen as a special case of distributed Bayesian computation where there is a natural way to merge information distributed in different locations. We will first describe this approach and then connect it to surrogate minimization to design better surrogates.

Consider a multitask setup in a Bayesian model where there are $t$ tasks each using a likelihood $p(\mathcal{D}_t|\boldsymbol{\theta})$ over data $\mathcal{D}_t$ and a common prior $p_0(\boldsymbol{\theta})$. For this case, there is a closed-form expression to quickly get the weighted multitask posterior. We first compute $T$ posteriors $p_t(\boldsymbol{\theta}) \propto p(\mathcal{D}_t|\boldsymbol{\theta})p_0(\boldsymbol{\theta})$, separately over their own likelihoods. The weighted posterior can be obtained by reusing the posterior $p_t$ by using the fact that the likelihood can be written as the ratio of posterior and prior $p(\mathcal{D}_t|\boldsymbol{\theta}) = p_t(\boldsymbol{\theta})/p_0(\boldsymbol{\theta})$. This is shown below,

$$p_{\boldsymbol{\alpha}}(\boldsymbol{\theta}) \;\propto\; p_0(\boldsymbol{\theta}) \prod_{t=1}^{T} p(\mathcal{D}_t|\boldsymbol{\theta})^{\alpha_t} \;\propto\; p_0(\boldsymbol{\theta})^\gamma \prod_{t=1}^{T} p_t(\boldsymbol{\theta})^{\alpha_t}. \quad (5)$$

The $\gamma = 1 - \sum_t \alpha_t$ is the same scalar used in front of the regularizer in Eqs. 2 and 3. Such *posterior merging* is a popular method in Bayesian literature, for example, see Bayesian committee machine (Tresp, 2000) or Bayesian data fusion (Mutambara, 1998; Durrant-Whyte & Stevens, 2001).

In fact, by choosing the Bayesian model appropriately, we can even exactly recover the solution for the weighted multitask problems. For example, suppose we want to recover the minimizer $\boldsymbol{\theta}_{\boldsymbol{\alpha}}$ by minimizing the objective $\gamma \mathcal{R}_0 + \sum_t \alpha_t \ell_t$, then we can choose

$$p(\mathcal{D}_t|\boldsymbol{\theta}) \propto \exp(-\ell_t(\boldsymbol{\theta})), \qquad p_0(\boldsymbol{\theta}) \propto \exp(-\mathcal{R}_0(\boldsymbol{\theta})).$$

These choices are valid within the generalized-Bayesian framework (Zhang, 1999; Catoni, 2007; Bissiri et al., 2016). With these, the minimizer $\boldsymbol{\theta}_{\boldsymbol{\alpha}}$ is simply the maximum-a-posterior (MAP) solution of the merged posterior $p_{\boldsymbol{\alpha}}$, that is,

$$\boldsymbol{\theta}_{\boldsymbol{\alpha}} = \arg\max_{\boldsymbol{\theta} \in \mathbb{R}^P} \log p_{\boldsymbol{\alpha}}(\boldsymbol{\theta}) = \arg\max_{\boldsymbol{\theta} \in \mathbb{R}^P} \gamma \underbrace{\log p_0(\boldsymbol{\theta})}_{=\widehat{\mathcal{R}}_0(\boldsymbol{\theta})} + \sum_{t=1}^{T} \alpha_t \underbrace{\log p_t(\boldsymbol{\theta})}_{=\widehat{\ell}_t(\boldsymbol{\theta})}. \quad (6)$$

Comparing this objective to Eqs. 2 and 3, we see that the Bayesian framework suggests to use the surrogate $\widehat{\ell}_t(\boldsymbol{\theta}) = -\log p_t(\boldsymbol{\theta})$ and regularizer $\widehat{\mathcal{R}}_0(\boldsymbol{\theta}) = -\log p_0(\boldsymbol{\theta})$. These surrogates are perfect because using them recovers the exact solution, but computing them exactly is also difficult. Our key idea is to use approximate Bayesian learning, specifically variational learning, to obtain posterior approximations and use them as surrogates.

## 3.3 Variational Bayesian Learning to Build Exponential-Family Surrogates

To build surrogates $\widehat{\ell}_t$ for each task, we propose to use variational learning which finds a posterior approximation $q_t(\boldsymbol{\theta})$ to the exact posterior $p_t(\boldsymbol{\theta})$,

$$q_t(\boldsymbol{\theta}) = \arg\min_{q \in \mathcal{Q}} \ \mathbb{E}_q[\ell_t(\boldsymbol{\theta})] + \mathbb{D}_{\mathrm{KL}}[q(\boldsymbol{\theta}) \,\|\, p_0(\boldsymbol{\theta})]. \tag{7}$$

We choose $\mathcal{Q}$ to be the set of exponential-family approximations or their mixtures, for example, Gaussian distribution. For such posteriors, we have good optimizers that perform well at large scale (Khan & Rue, 2023). For instance, for Gaussians, the above problem can be optimized by using Adam-like optimizers (Shen et al., 2024). Even Adam can be seen as solving this problem (Khan et al., 2018). We can just use such optimizers to compute the posterior $q_t$.

Motivated by Eq. 6, we propose to use $q_t(\boldsymbol{\theta})$ to build the surrogate and estimate $\boldsymbol{\theta_\alpha}$, as shown below:

$$\widehat{\ell}_t(\boldsymbol{\theta}) = -\log q_t(\boldsymbol{\theta}) \quad \Longrightarrow \quad \widehat{\boldsymbol{\theta}_\alpha} = \arg\max_{\boldsymbol{\theta}} \ \gamma \widehat{\mathcal{R}}_0(\boldsymbol{\theta}) + \sum_{t=1}^{T} \alpha_t \widehat{\ell}_t(\boldsymbol{\theta}) \tag{8}$$

As for the prior $\widehat{\mathcal{R}}_0$, we will use a Gaussian prior, although other choices are also possible. These choices can recover existing model-merging strategies. For example, we recover Eq. 2 up to a constant by choosing $q_t$ to be the Gaussian approximation shown below:

$$q_t(\boldsymbol{\theta}) = \mathcal{N}(\boldsymbol{\theta} \,|\, \boldsymbol{\theta}_t, \mathbf{I}) \quad \Longrightarrow \quad \widehat{\ell}_t(\boldsymbol{\theta}) = \tfrac{1}{2}\|\boldsymbol{\theta} - \boldsymbol{\theta}_t\|^2 + \mathrm{const}. \tag{9}$$

The posterior $q_t(\boldsymbol{\theta})$ is a Laplace approximation obtained as a special case of variational learning (Khan & Rue, 2023, Eq. 25). The prior can be chosen to be isotropic Gaussian: $p_0(\boldsymbol{\theta}) = \mathcal{N}(\boldsymbol{\theta} \,|\, 0, \mathbf{I})$. Similarly, Task Arithmetic can be derived just by simply changing the prior to $p_0(\boldsymbol{\theta}) = \mathcal{N}(\boldsymbol{\theta} \,|\, \boldsymbol{\theta}_{\mathrm{LLM}}, \mathbf{I})$.

Next, Hessian-Weighted merging methods are obtained by using a full-Gaussian posterior, again with the Laplace approximation $q_t(\boldsymbol{\theta}) = \mathcal{N}(\boldsymbol{\theta} \,|\, \boldsymbol{\theta}_t, \mathbf{H}_t^{-1})$ using Hessian $\mathbf{H}_t$. With this, we get the squared Mahalanobis distance $\widehat{\ell}_t(\boldsymbol{\theta}) = \tfrac{1}{2}\|\boldsymbol{\theta} - \boldsymbol{\theta}_t\|^2_{\mathbf{H}_t}$ as the surrogate, which yields a Hessian-Weighted merging (detailed derivation is included in App. A.1),

$$\widehat{\boldsymbol{\theta}_\alpha}^{\mathrm{Hess}} = \Big( \sum_t \alpha_t \mathbf{H}_t \Big)^{-1} \sum_t \alpha_t \mathbf{H}_t \boldsymbol{\theta}_t. \tag{10}$$

Here, we use $p_0(\boldsymbol{\theta}) = \mathcal{N}(\boldsymbol{\theta} \,|\, 0, \mathbf{I})$ and $\sum_t \alpha_t = 1$. If we make similar choices for the prior as we did in the Task Arithmetic case, we recover the method proposed by Daheim et al. (2024).

In general, we can use any exponential-family posterior and employ them as surrogates. Such surrogates always have a closed-form solution. This is because of the form of the posterior,

$$q_t(\boldsymbol{\theta}) \propto e^{\boldsymbol{\lambda}_t^\top \mathbf{T}(\boldsymbol{\theta})} \quad \Longrightarrow \quad \widehat{\ell}_t(\boldsymbol{\theta}) = -\boldsymbol{\lambda}_t^\top \mathbf{T}(\boldsymbol{\theta}) + \mathrm{const}.$$

where we denote sufficient statistics by $\mathbf{T}(\boldsymbol{\theta})$ and natural parameter by $\boldsymbol{\lambda}_t$. For example, for Gaussian, we have $\mathbf{T}(\boldsymbol{\theta}) = (\boldsymbol{\theta}, \boldsymbol{\theta}\boldsymbol{\theta}^\top)$ giving rise to the quadratic surrogates derived earlier. The merged model $\boldsymbol{\theta_\alpha}$ obtained by solving Eq. 8 always has a closed-form because it is equivalent to the MAP of an exponential family distribution. This is shown in App. A.2. The surrogates not only take flexible forms, but are also globally accurate. This is because they are obtained by solving Eq. 7 which is equivalent to minimizing the KL divergence to the exact posterior $p_t$. Minimizing the divergence ensures that the surrogates are accurate not only locally at $\boldsymbol{\theta}_t$ but also globally in regions where $q_t$ has high probability mass; see a discussion in (Opper & Archambeau, 2009).

## 3.4 Improved Merging via Mixtures of Exponential-Families

Here, we extend to mixture of exponential-family distributions which provide more expressive posteriors and therefore even more accurate surrogates. For simplicity, we assume that $\sum_t \alpha_t = 1$,

---

**Algorithm 1** Fast and cheap multitask previews via mixture of Gaussian merging

---

**Require:** $K$ different Gaussians for each of the $T$ tasks $\mathcal{N}(\boldsymbol{\theta} \mid \boldsymbol{\theta}_{tk}, \text{diag}(1/\mathbf{h}_{tk}))$.
 1: **for** all $\boldsymbol{\alpha}$ values in the preview **do**
 2:     Initialize $\widehat{\boldsymbol{\theta}}_{\boldsymbol{\alpha}}$ and set $\pi_k = 1/K$ for all $k$.
 3:     **while** not converged **do**
 4:         For all $t, k$: compute $\tilde{p}_{tk} = \pi_k \mathcal{N}(\widehat{\boldsymbol{\theta}}_{\boldsymbol{\alpha}} \mid \boldsymbol{\theta}_{tk}, \text{diag}(1/\mathbf{h}_{tk}))$; normalize $\hat{\pi}_{tk} \leftarrow \frac{p_{tk}}{\sum_{k'} p_{tk'}}$
 5:         $\mathbf{h}_{\boldsymbol{\alpha}} \leftarrow \sum_{t,k} \hat{\pi}_{tk} \alpha_t \mathbf{h}_{tk}$
 6:         $\widehat{\boldsymbol{\theta}}_{\boldsymbol{\alpha}} \leftarrow (\sum_{t,k} \hat{\pi}_{tk} \alpha_t \mathbf{h}_{tk} \boldsymbol{\theta}_{tk})/\mathbf{h}_{\boldsymbol{\alpha}}$
 7:     **end while**
 8: **end for**

---

so there is no regularizer. While mode finding for mixtures is still tractable, it requires an iterative expectation-maximization (EM) procedure which should still be cheap if it converges within a few steps. We assume that the $k$'th mixture component is an EF with natural parameter $\boldsymbol{\lambda}_{tk}$. Each component is weighted by $\pi_k > 0$ and $\sum_k \pi_k = 1$. Then, the posterior and surrogate take the following form:

$$q_t \propto \sum_{k=1}^{K} \pi_k \underbrace{e^{\boldsymbol{\lambda}_{tk}^\top \mathbf{T}(\boldsymbol{\theta})}}_{\propto \tilde{p}_{tk}(\boldsymbol{\theta})} \quad \implies \quad \widehat{\ell}_t(\boldsymbol{\theta}) = -\log \sum_{k=1}^{K} \pi_k e^{\boldsymbol{\lambda}_{tk}^\top \mathbf{T}(\boldsymbol{\theta})}. \tag{11}$$

Clearly, the surrogate is much more expressive than quadratics surrogates used in model merging.

Despite the non-concavity of the objective, we can maximize it using an iterative Expectation-Maximization (EM) approach where each step has a closed-form solution. A detailed derivation is in App. A.3. As a special case, consider mixture-of-Gaussians (MoG) posterior where the updates take the following form similarly to Eq. 10:

$$\boldsymbol{\theta}^{(i+1)} \leftarrow (\mathbf{H}_{\boldsymbol{\alpha}}^{(i)})^{-1} \sum_{t,k} \hat{\pi}_{tk}^{(i)} \alpha_t \mathbf{H}_{tk} \boldsymbol{\theta}_{tk}, \text{ where } \mathbf{H}_{\boldsymbol{\alpha}}^{(i)} = \sum_{t,k} \hat{\pi}_{tk}^{(i)} \alpha_t \mathbf{H}_{tk}. \tag{12}$$

The main difference is that each component is now weighted by $\hat{\pi}_{tk}^{(i)} \propto \pi_k \mathcal{N}(\boldsymbol{\theta}^{(i)} \mid \boldsymbol{\theta}_{tk}, \mathbf{H}_{tk}^{-1})$, normalized over $k$. This update generalizes the fixed-point algorithm of Carreira-Perpiñán (2000, Section 5) which was proposed to find the modes of Gaussian mixtures.

### 3.5 PRACTICAL ALGORITHMS FOR FAST MULTITASK FINETUNING PREVIEWS

Our methods to generate previews follow the 3-step procedure described in Sec. 3. Below we list three versions where different algorithms are used to generate $q_t(\boldsymbol{\theta})$.

1. ADAMW-SG: We train each task using AdamW (Loshchilov & Hutter, 2019) to get $\boldsymbol{\theta}_t$ and use it to build the Laplace posterior $q_t(\boldsymbol{\theta}) = \mathcal{N}(\boldsymbol{\theta} \mid \boldsymbol{\theta}_t, \mathbf{H}_t^{-1})$. The Hessian is fixed to a diagonal matrix with the diagonal set to squared gradients, which are computed by one extra pass through the data. Previews are then computed using Eq. 10 for all $\boldsymbol{\alpha}$.

2. IVON-HESS: We train each task using the variational learning method IVON (Shen et al., 2024) which yields a Gaussian with diagonal covariance, similarly to AdamW-SG. The advantage here is that no additional pass through the data is required to compute the diagonal Hessian. Again, previews are computed by using Eq. 10.

3. MULTIIVON-HESS. For each task, we train using multiple runs of IVON and use them to construct a mixture-of-Gaussians posterior. The cost is $K$ times the cost of IVON-Hess where $K$ is the number of IVON runs. Previews are obtained using Alg. 1.

We also show results for using full Hessians on small-scale experiments in Fig. 2 and App. C.1; details are given in App. C.1 and Appendices B.1 and B.2. For MultiIVON-Hess, we only run 5-10 iterations of Alg. 1 with $K$ set in the range of 10 to 30 components. The method is still practical but, compared to other methods, this can have larger training overhead for large models due to having to train 10-30 models for each task.

|  | Figure | Model | Tasks | Simple Merging | Hessian Weighted | Mixture Weighted |
|---|---|---|---|---|---|---|
| CV | Fig. 8 | Logistic | MNIST Imbalanced | 0.2067 | 0.1528 | **0.0463** |
|  | Fig. 9 | Logistic | MNIST Balanced | 0.2015 | 0.1191 | **0.0809** |
|  | Fig. 3 | ResNet-20 | CIFAR-10 | 0.0252 | 0.0098 | **0.0084** |
|  | Fig. 4 | ViT-B/32 | RESISC45, GTSRB, SVHN | 0.0388 | **0.0263** | - |
|  |  |  | EuroSAT, Cars, SUN397 | 0.0084 | **0.0059** | - |
| NLP | Fig. 6 | RoBERTa | RT, SST-2, Yelp | 0.0314 | **0.0281** | - |
|  | Fig. 7 | GEMMA-2B | IWSLT2017de-en, fr-en | 0.5380 | **0.4657** | - |

Table 1: MSE between performance metrics of models obtained with multitask finetuning and models obtained with model merging for various task weightings. Model merging methods with better posterior approximations more closely match the multitask finetuned models' performances.

|  | Figure | Model | Simple Merging | Hessian Weighted | Mixture Weighted | Multitask Finetuning |
|---|---|---|---|---|---|---|
| CV | Fig. 8 | Logistic | 76.8% 90.0% | 80.3% 89.4% | 87.2% 90.4% | 90.4% |
|  | Fig. 9 | Logistic | 71.3% 90.7% | 78.5% 90.5% | 83.2% 90.2% | 90.7% |
|  | Fig. 3 | ResNet-20 | 62.4% 67.4% | 62.3% 64.6% | 64.5% 67.4% | 68.1% |
|  | Fig. 4 | ViT-B/32 | 86.0% 97.1% | 88.3% 97.1% | - | 97.3% |
|  |  |  | 79.2% 83.9% | 80.0% 84.0% | - | 84.5% |
| NLP | Fig. 6 | RoBERTa | 93.5% 95.2% | 93.6% 95.3% | - | 95.5% |
|  | Fig. 7 | GEMMA-2B | 25.9 26.4 | 25.9 26.7 | - | 26.7 |

Table 2: We report maximum accuracy/BLEU obtained via previews; the corresponding $\alpha$ values are reported in Table 3 in the appendix. With each score, we also show the true scores (shown in gray) of multitask finetuning obtained using the same weights. Finally, the maximum scores for multitask finetuning over all $\alpha$ values are shown in the last column. We see a consistent trend that the performance improves as we use better posteriors. For example, for Fig. 8, the accuracy increases from 76.8% to 87.2% which is quite close to the best performance of 90.4% for multitask finetuning.

## 4 EXPERIMENTS & RESULTS

We compare multitask finetuning and model merging on image classification using ResNets (Sec. 4.1) and vision transformers (Sec. 4.2), text classification with masked language models (Sec. 4.3), and machine translation using LLMs (Sec. 4.4). In App. C.1 we explore model merging on logistic regression. Overall, better posterior approximations match multitask finetuning more closely both in terms of overall shape of the solution (Table 1) and the performance of the best weightings (Table 2).

### 4.1 IMAGE CLASSIFICATION ON CIFAR-10

Next, we move to a neural network finetuning. We first pretrain ResNet-20 (He et al., 2016) with 260k parameters on a subset of CIFAR-10 and then finetune this checkpoint on the remaining examples. The finetuning tasks are: (1) airplane, car, ship, truck; (2) cat, dog; (3) deer, dog, frog, horse. We compare Hessian-Weighted (IVON-HESS) and Mixture-Weighted (MULTIIVON-HESS) to simple averaging and Exact Solution (Joint training). Hyperparameters are in App. B.3.

Results are shown in Fig. 3 and again show that better posterior approximations yield better previews. For example, Simple Merging misses that performance is still good in any of the corners, especially the top and bottom right one. Hessian-Weighted merging misses the best-performing region and would suggest exploring a region slightly above it. Mixture-Weighted previews are much better and improve with the number of components. Interestingly, the best-performing region of this approach moves further down and right, that is, closer to that of the joint training solution.

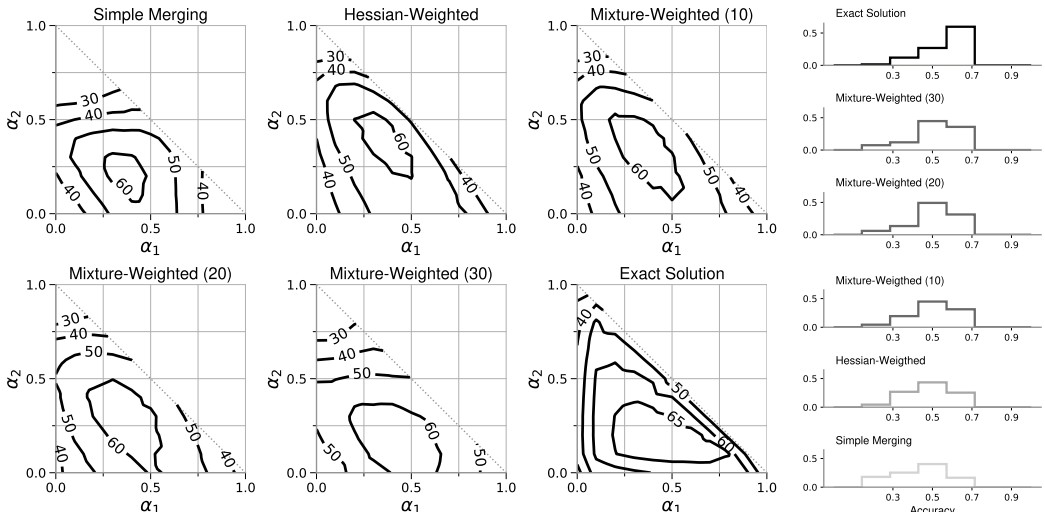

Figure 3: Results on image classification using ResNet-20 on CIFAR-10 with three tasks constructed from different sets of classes. Preview quality improves with the expressiveness of the posterior approximation. Notably, more mixture components improve the preview. Hessian-Weighted previews generated with IVON-HESS and Mixture-Weighted with MULTIIVON-HESS. Histograms show that the distribution of weights that achieve a similar accuracy also improves with better posteriors.

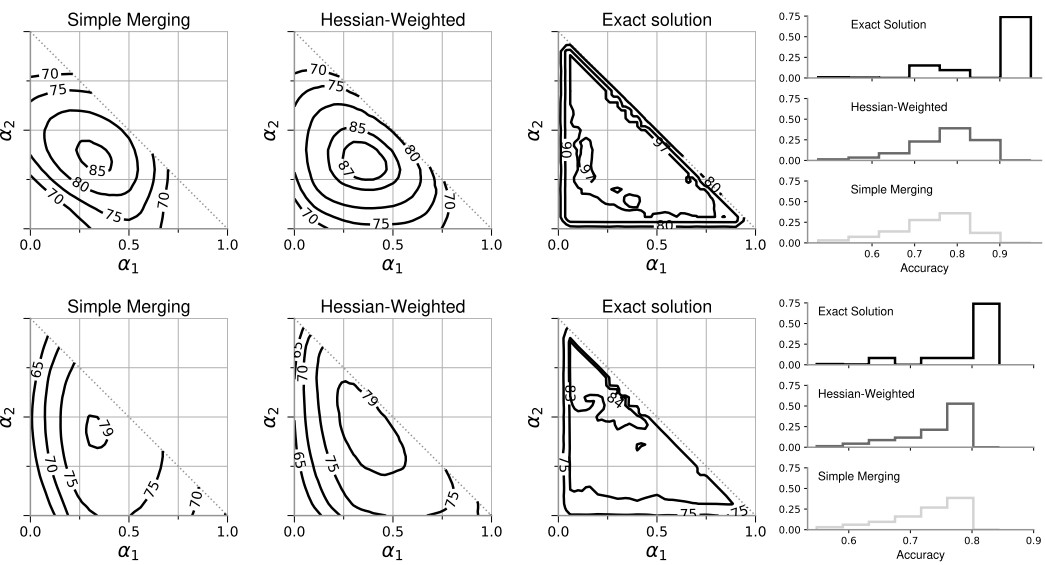

Figure 4: Results using ViT-B/32 on GTSRB, RESISC45, SVHN (top) and EuroSAT, Cars, Sun397 (bottom). The exact solution shows a large triangular area of well-performing weightings which is better captured by Hessian-Weighted merging. Simple Merging especially fails around the edges, whereas Hessian-Weighted (ADAMW-SG) performs much better (right). Similarly we see on the histograms that the Hessian uncovers more high accuracy weights than the Simple Merging.

## 4.2 VISION TRANSFORMERS

Next, we experiment with multitask finetuning ViT-B/32 models (Dosovitskiy et al., 2021) based on CLIP (Radford et al., 2021) for image classification. First, we use GTSRB (Houben et al., 2013), RESISC45 (Cheng et al., 2017) and SVHN (Netzer et al., 2011); then we use EuroSAT (Helber et al., 2019), Stanford Cars (Krause et al., 2013) and SUN397 (Xiao et al., 2010). We compare Simple

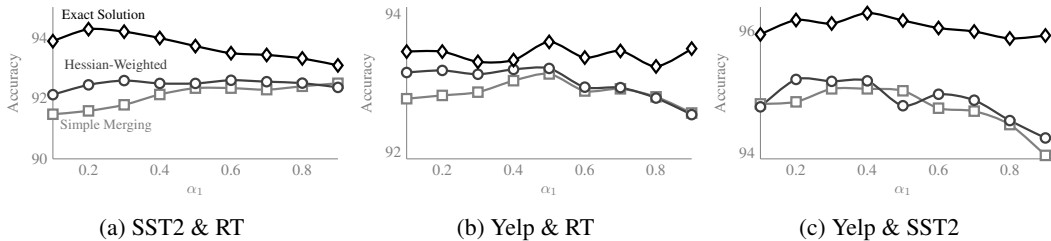

(a) SST2 & RT  (b) Yelp & RT  (c) Yelp & SST2

Figure 5: Merging of multitask finetuned RoBERTa models on pairs of sentiment analysis tasks. Model merging provides good previews of weightings for multitask finetuning but some trends (e.g. $\alpha_1 \in [0.0, 0.5]$ for SST2&RT) are only picked up by better posteriors and Hessian-Weighted merging (ADAMW-SG). First-named task is weighted by $\alpha_1$ and the other by $1 - \alpha_1$.

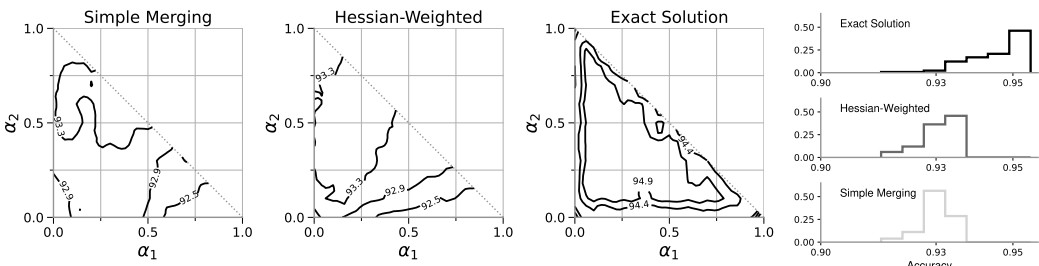

Figure 6: Preview of reweightings for RoBERTA multitask finetuned on three sentiment analysis tasks. Hessian-Weighting achieves a better preview by showing a larger region of performant weightings. The histogram shows that it more accurately recovers the trend of the weightings.

Merging to Hessian-Weighted (ADAMW-SG) and provide. Further details on training and evaluation in App. B.4. The joint solution uses a grid with spacing $0.05$ to explore the possible sets of $\boldsymbol{\alpha}$.

The results are shown in Fig. 4. In both cases, the joint solution is similar and shows that almost all weightings that are not directly at the borders (where one of the tasks gets a very small weight) have good performance. While for EuroSAT, Cars, SUN397 there is a smaller region with accuracy over 84 points in accuracy, any weightings in the large contour of above 80 points will still be good. Ideally this should be reflected by a preview from merging. Hessian-based merging shows the flat triangular shape of the joint solution better than the simple method. Training times highlight the usefulness: joint training for one weighted combination takes around 51 minutes while merging takes only seconds (plus 14-17 minutes for each finetuning on separate tasks).

## 4.3 MASKED LANGUAGE MODELS

In this section, we show results when multitask finetuning masked language models for text classification. We follow Daheim et al. (2024) and train RoBERTa (Liu et al., 2019) first on the IMDB sentiment classification task Maas et al. (2011). Then, we finetune on Rotten Tomatoes (RT) (Pang & Lee, 2005), SST-2 (Socher et al., 2013), and Yelp (Zhang et al., 2015), and merge the resulting models. We use two settings: the first merges all combinations of two of the three finetuned models; the second merges all three. Due to heavy compute requirements, we run joint training with a coarser grid than model merging in the latter. We compare Simple Merging and Hessian-Weighted (ADAMW-SG).

Results for merging two models are shown in Fig. 5 where we see that even simple merging can often produce good previews but fails for specific weightings. For example, on SST2 and RT the best-performing factors for Simple Merging ($\alpha_1 = 0.9$) are the worst-performing in the joint solution. Using a diagonal Gaussian instead of an isotropic one shows a more similar trend to this joint solution. The results for merging all three tasks are shown in Fig. 6. Here, the exact solution again shows a fairly triangular shape but this is not at all reflected in the simpler merging scheme.

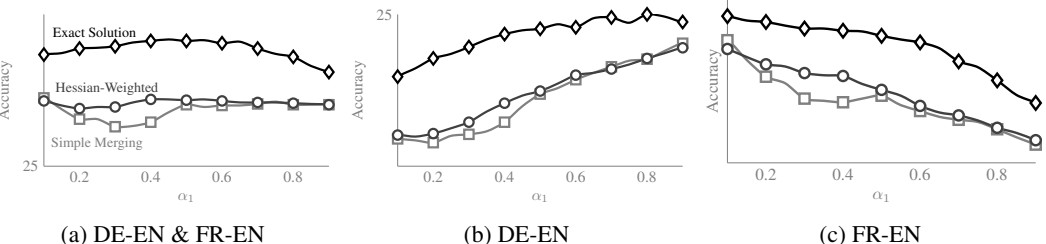

(a) DE-EN & FR-EN        (b) DE-EN        (c) FR-EN

Figure 7: Here, we merge LoRA-finetuned GEMMA-2B models trained on IWSLT2017de-en and IWSLT2017fr-en. Again, model merging can faithfully preview the trends of joint training with different weights. For this, better posterior approximations (Hessian-Weighted with IVON-HESS) are important, for example, around $\alpha = 0.4$ on IWSLT2017fr-en.

### 4.4 MACHINE TRANSLATION WITH FINETUNED LLMS

Next, we show that our methods apply to LLMs with more than one billion parameters, also if they are finetuned using parameter-efficient finetuning strategies such as LoRA. In particular, we merge two GEMMA-2B-it (Gemma Team, 2024b) models finetuned on IWSLT2017 (Cettolo et al., 2017) de-en and fr-en, respectively, and compare them to training jointly on both language pairs. We use IVON-HESS for Hessian-Weighted merging. Details about the experimental set-up are in App. B.6.

Results are shown in Fig. 7. There, we find that Simple Merging does not always match the shape of the joint training solution, especially around $\alpha = 0.4$. Using Hessian-Weighted merging improves this. Overall, this shows that the our method can also scale to larger models and datasets, even if only a small subset of the parameters is adapted during finetuning. One run of multitask finetuning for a specific weight takes around 17 hours while merging takes just 1 minute (plus 8-9 hours for finetuning on each task separately).

## 5 CONCLUSION

Multitask finetuning is a crucial ingredient in many neural network training recipes but good weightings between tasks are hard and expensive to find. Here, we propose to aid the search for such weightings with previews obtained from model merging, where single task models can be reused for many weight combinations. We show that model merging strategies can be derived using a Bayesian framework by defining suitable surrogate losses to the multitask objective for exponential-family-based distributions. We use this to outline various preview and merging strategies, including a new mixture-based algorithm for improved model merging. Along various experiments including image classification with Vision Transformers and machine translation with LLMs we show that model merging can effectively be used to preview multitask finetuning weightings. Flexible model merging can improve the preview quality, but also increase the cost. For example, mixture posteriors sometimes need too many mixture component to get improvements. This is not ideal for large model where storing many models is not possible. We hope to address this limitation in the future.

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

# A    DERIVATIONS

## A.1    DERIVATION OF HESSIAN-WEIGHTED MERGING

For simplicity, we set $\sum_t \alpha_t = 1$, which implies that $\gamma = 0$ (the derivation can easily be extended to other cases). For $q_t(\boldsymbol{\theta}) = \mathcal{N}(\boldsymbol{\theta} \mid \boldsymbol{\theta}_t, \mathbf{H}_t^{-1})$, the problem in Eq. 8 can be written as

$$
\begin{aligned}
\widehat{\boldsymbol{\theta}}_{\boldsymbol{\alpha}} &= \arg\min_{\boldsymbol{\theta}} \ \gamma\widehat{\mathcal{R}}_0(\boldsymbol{\theta}) + \sum_{t=1}^{T} \alpha_t \widehat{\ell}_t(\boldsymbol{\theta}) \\
&= \arg\min_{\boldsymbol{\theta}} \ \sum_{t=1}^{T} \alpha_t \tfrac{1}{2}\|\boldsymbol{\theta} - \boldsymbol{\theta}_t\|_{\mathbf{H}_t}^2 \\
&= \arg\min_{\boldsymbol{\theta}} \ \boldsymbol{\theta}^\top \left( \sum_{t=1}^{T} \alpha_t \mathbf{H}_t \right) \boldsymbol{\theta} - \boldsymbol{\theta}^\top \left( \sum_{t=1}^{T} \alpha_t \mathbf{H}_t \boldsymbol{\theta}_t \right)
\end{aligned}
$$

Minimizing this, we get Eq. 10.

## A.2 Closed-Form Expression for MAP of Exponential-Family Distribution

For simplicity, we will ignore the prior $p_0$, but the derivation can be straightforwardly extended to the case when it is included. We start by noting that minimizing $\sum_t \alpha_t \widehat{\ell}_t$ is equivalent finding MAP of $\prod_t q_t(\boldsymbol{\theta})^{\alpha_t}$. Let us denote it by $p_{\boldsymbol{\alpha}}$. The mode of $p_{\boldsymbol{\alpha}}$ is available in closed-form because we can always rewrite the posterior in an exponential form that allows us to compute the max. This is shown below where we first rewrite the posterior with a log-partition function $A(\boldsymbol{\theta})$ and an alternate sufficient statistic $\mathbf{t}(\boldsymbol{\lambda}_\alpha)$, and then simply take the derivative to get a closed-form expression for $\widehat{\boldsymbol{\theta}}_{\boldsymbol{\alpha}}$,

$$p_{\boldsymbol{\alpha}} \propto \exp(\boldsymbol{\theta}^\top \mathbf{t}(\boldsymbol{\lambda}_\alpha) - A(\boldsymbol{\theta})) \implies \widehat{\boldsymbol{\theta}}_{\boldsymbol{\alpha}} = \arg\max_{\boldsymbol{\theta}} \log p_\alpha(\boldsymbol{\theta}) = \nabla A^{-1}\left(\mathbf{t}(\boldsymbol{\lambda}_\alpha)\right). \quad (13)$$

The alternate form is essentially following the form of the conjugate prior (see Bishop (1995, Eq. 2.229)) written to match the form of the likelihood. For instance, for Gaussian $p_{\boldsymbol{\alpha}} = \mathcal{N}(\boldsymbol{\theta}|\mathbf{m}_\alpha, \mathbf{H}_\alpha^{-1})$ obtained by, Hessian-Weighted merging in Eq. 10,

$$p_{\boldsymbol{\alpha}} \propto \exp(\boldsymbol{\theta}^\top \underbrace{\mathbf{H}_\alpha \mathbf{m}_\alpha}_{\mathbf{t}(\boldsymbol{\lambda}_\alpha)} - \underbrace{\tfrac{1}{2}\boldsymbol{\theta}^\top \mathbf{H}_\alpha \boldsymbol{\theta}}_{A(\boldsymbol{\theta})}) \implies \underbrace{\mathbf{H}_\alpha(\boldsymbol{\theta}_\alpha)}_{\nabla A(\boldsymbol{\theta}_\alpha)} = \underbrace{\mathbf{H}_\alpha \mathbf{m}_\alpha}_{\mathbf{t}(\boldsymbol{\lambda}_\alpha)} \implies \boldsymbol{\theta}_{\boldsymbol{\alpha}} = \mathbf{m}_\alpha$$

Solutions of variational objectives always have this form (Khan & Rue, 2023, Sec. 5), while convexity of $A(\boldsymbol{\theta})$ ensures that the mode always exists and can be easily found without retraining on individual objectives. In summary, we can always get a closed-form expression as follows

1. Compute $\boldsymbol{\lambda}_t$ of individual objectives.
2. Aggregate them $\boldsymbol{\lambda}_\alpha = \sum_t \alpha_t \boldsymbol{\lambda}_t$.
3. Compute $\boldsymbol{\theta}_{\boldsymbol{\alpha}}$ using Eq. 13.

Below, we discuss another example of the Beta-Bernoulli model to model coin-flips $y_t \in \{0, 1\}$ with probability $\pi$,

$$p(y_t \,|\, \pi) \propto \pi^{y_t}(1-\pi)^{1-y_t}, \quad p(\pi) \propto \pi^{a_0-1}(1-\pi)^{b_0-1}.$$

The unknown is then modeled as $\theta = \log(\pi/(1-\pi))$, to get the posterior which is also a Beta distribution with parameters

$$a_t = y_t + a_0, \quad b_t = 1 - y_t + b_0, \quad p(\theta \,|\, \mathcal{D}_t) \propto \pi^{a_t-1}(1-\pi)^{b_t-1} \propto \exp(\boldsymbol{\lambda}_t^\top \mathbf{T}(\theta))$$

where $\mathbf{T}(\theta) = (\log \pi, \log(1-\pi))$ and $\boldsymbol{\lambda}_t = (a_t - 1, b_t - 1)$. With this, the aggregate posterior $p_\alpha$ is a Beta distribution natural parameter $\boldsymbol{\lambda}_\alpha = (a_\alpha - 1, b_\alpha - 1)$ where $a_\alpha$ and $b_\alpha$ are simply a weight average of $a_t$ and $b_t$ respectively. To get the maximum of $p_\alpha$, we write it in an exponential form,

$$p_\alpha(\theta) \propto \exp(\theta \underbrace{a_\alpha - 1}_{t(\boldsymbol{\lambda}_\alpha)} - \underbrace{(a_\alpha + b_\alpha - 2)\log(1 + e^\theta)}_{A(\theta)})$$

We can then compute $\nabla A(\boldsymbol{\theta})$, plug it in Eq. 13, and simplify to get the closed-form update:

$$\underbrace{\frac{a_\alpha + b_\alpha - 2}{1 + e^{-\hat{\theta}_\alpha}}}_{\nabla A(\hat{\theta}_\alpha)} = \underbrace{a_\alpha - 1}_{\mathbf{t}(\boldsymbol{\lambda}_\alpha)} \implies \hat{\theta}_\alpha = \underbrace{\log \frac{a_\alpha - 1}{b_\alpha - 1}}_{\nabla A^{-1}(\mathbf{t}(\boldsymbol{\lambda}_\alpha))} \implies \hat{\pi}_\alpha = \log \frac{a_\alpha - 1}{a_\alpha + b_\alpha - 2}.$$

## A.3 Derivation of the EM algorithm for Mixture Posteriors

To do so, we use the EM algorithm by viewing the summation over $k$ in Eq. 11 as marginalization over a discrete variable $z_k \in \{1, 2, \ldots, K\}$ of the joint $p(\boldsymbol{\theta}, z_t = k) = p_{tk}(\boldsymbol{\theta})$. Then, given parameters $\boldsymbol{\theta}^{(i)}$ at each iteration $i$, we maximize the EM lower bound. The posterior over $z_k$ is

$$p(z_t = k \,|\, \boldsymbol{\theta}^{(i)}) = \hat{\pi}_{tk}^{(i)} = \frac{p_{tk}(\boldsymbol{\theta}^{(i)})}{\sum_{k'} p_{tk'}(\boldsymbol{\theta}^{(i)})}.$$

Using this, we can write the following lower bound,

$$\sum_{t=1}^T \alpha_t \log \left( \sum_{k=1}^K \frac{p_{tk}(\boldsymbol{\theta})}{\hat{\pi}_{tk}^{(i)}} \hat{\pi}_{tk}^{(i)} \right) \geq \sum_{t=1}^T \sum_{k=1}^K \alpha_t \hat{\pi}_{tk}^{(i)} \log p_{tk}(\boldsymbol{\theta}) + \mathrm{c} = \underbrace{\sum_{t=1}^T \sum_{k=1}^K \alpha_t \hat{\pi}_{tk}^{(i)} \boldsymbol{\lambda}_{tk}^\top}_{=\boldsymbol{\lambda}_\alpha^{(i)}} \mathbf{T}(\boldsymbol{\theta}) + \mathrm{c}.$$

The above corresponds to the log of an exponential family (denoted by $p_\alpha^{(i)}$) with natural parameter $\boldsymbol{\lambda}_\alpha^{(i)}$, which gives us the following iterative procedure:

$$\boldsymbol{\theta}^{(i+1)} \leftarrow \arg\max_{\boldsymbol{\theta}} \sum_{t=1}^{T} \sum_{k=1}^{K} \hat{\pi}_{tk}^{(i)} \alpha_t \boldsymbol{\lambda}_{tk}^{\top} \mathbf{T}(\boldsymbol{\theta}), \tag{14}$$

where we use the posterior $\hat{\pi}_{tk}^{(i)} = p(z_t = k \,|\, \boldsymbol{\theta}^{(i)}) \propto p_{tk}(\boldsymbol{\theta}^{(i)})$ which is obtained by normalizing over $k$. The iterates $\boldsymbol{\theta}^{(i)}$ converge to a local maximum which provides a solution for $\hat{\boldsymbol{\theta}}_\alpha$. For $K = 1$, the algorithm reduces to the exponential-family case.

# B EXPERIMENTAL SETUP

## B.1 ILLUSTRATIVE 2D EXAMPLE

The individual functions in Fig. 2 are of the form $\ell_t(\boldsymbol{\theta}) = \log\left(\sum_{i=1}^{N} \exp(\mathbf{a}_{it}^{\top} \boldsymbol{\theta} + b_{it})\right)$ where $\mathbf{a}_{it}, b_{it}$ are chosen randomly from normal distributions for $t = 1, 2$ and uniformly for $t = 3$. We approximate the Gibbs distributions $\exp(-\ell_t(\boldsymbol{\theta}))$ using the mixture-of-Gaussian algorithm described in Lin et al. (2019, Section 4.1) with full Hessians, and we use the EM algorithm described in Eq. 12 to find the mode of the mixture.

## B.2 MERGING LOGISTIC REGRESSION MODELS

For parameter averaging we train each model using gradient-descent with learning-rate $\rho = 3.0$ for 2500 iterations, and use $\sum_t \alpha_t \boldsymbol{\theta}_t$ to obtain each $\hat{\boldsymbol{\theta}}_\alpha$. For the full-Gaussian method, which we use for Hessian-weighted merging, we implement the variational online Newton method described in Khan & Rue (2023, Section 1.3.2). We set the learning-rate $\rho_t = 0.1$, perform 3 Monte-Carlo samples to estimate the expected gradient and Hessian and run for 25 iterations. The parameters of the merged model are obtained via Eq. 10. The mixture of full-Gaussian trains each model by the method described in Lin et al. (2019, Section 4.1) with a 20 component mixture. We set the algorithm's learning-rate of $\beta = 0.02$ for the mean $\boldsymbol{\mu}$ and precision $\boldsymbol{\Sigma}^{-1}$, $\beta = 3 \times 10^{-6}$ for the mixture weights $\pi$, while Monte-Carlo samples number of iterations are the same as full-Gaussian. The test-accuracy in Fig. 8 and Fig. 9 is plotted on a grid $\boldsymbol{\Delta}$ with uniform spacing 0.02. The tasks are for imbalanced (T1: $\{0, 1\}$, T2: $\{2, 3, 4\}$ and T3: $\{5, 6, 7, 8, 9\}$); and for balanced: (T1: $\{0, 1, 2\}$, T2: $\{3, 4, 5, 6\}$, T3: $\{7, 8, 9\}\}$).

## B.3 MERGING VISION MODELS ON CIFAR-10

We pretrain the ResNet-20 model by running the IVON optimizer for 1000 epochs with 5 Monte-Carlo samples to estimate expected gradients and Hessians and use IVON-HESS for previews. The hyperparmeters of IVON are set as follows: learning-rate $\alpha = 0.1$, momentum $\beta = (0.9, 0.9999)$, weight-decay $\delta = 10^{-3}$ and temperature/sample-size weighting $\lambda = 50000$. The batch-size is set to 50 and the estimated Hessian is initialized to 0.1.

The individual models are also finetuned with IVON, initialized at the pretrained posterior for $\{25, 50, 75, 100, 125, 150\}$ steps over 5 random seeds to obtain a soup of 30 models for each task. This corresponds to the MULTIIVON-HESS method. Each step processes 1000 examples, where the batch-size is set to 50. No weight-decay is used for finetuning, and we use a smaller learning-rate $\alpha = 0.01$. For the batch solution, we finetune on all data for 250 steps with the same hyperparameters.

The merged models $\hat{\boldsymbol{\theta}}_\alpha$ are computed using Alg. 1, where we take the models across 5 random seeds with $\{100, 150\}$, $\{75, 100, 125, 150\}$ and $\{25, 50, 75, 100, 125, 150\}$ steps for the mixtures of size 10, 20 and 30. The test-accuracy in Fig. 3 is plotted on a grid $\boldsymbol{\Delta}$ with uniform spacing 0.1.

## B.4 MERGING VISION TRANSFORMERS

The pretrained and finetuned checkpoints of ViT-B-32 a model based on CLIP (Radford et al., 2021) on these downstream tasks (RESISC45 (Cheng et al., 2017), GTSRB (Houben et al., 2013), SVHN (Netzer et al., 2011), EuroSAT (Helber et al., 2019), Stanford Cars (Krause et al., 2013)

and SUN397 (Xiao et al., 2010)) were obtained based on the code from `https://github.com/mlfoundations/task_vectors`. The squared-gradients approximation for the Hessian-Weighted merge with ADAMW-SG is computed by $\sum_i \nabla \ell_i(\boldsymbol{\theta}_t)^2$, where $i$ is a sum over data examples from the training data.

To generate the exact solution contours we start from the pretrained checkpoint and finetune on the joint datasets with weights obtained by sampling from a grid $\boldsymbol{\Delta}$ with spacing $0.05$. The optimizer is AdamW, with learning rate of $10^{-5}$, set $(\beta_1, \beta_2) = (0.9, 0.999)$ and decay the learning rate to $0$ using a cosine decay with 500 warmup steps. Training is done for 15 epochs on RESISC45,GTSRB and SVHN, while for EuroSAT, Stanford Cars and SUN397 this was set to 35, in both experiments batch size is 128.

### B.5 MERGING MASKED LANGUAGE MODELS

We pretrain RoBERTa with 125M parameters using AdamW on the IMDB dataset for sentiment classification (Maas et al., 2011). We use a learning rate of $10^{-5}$ and set $(\beta_1, \beta_2) = (0.9, 0.999)$ and decay the learning rate to $0$ using a cosine decay with 100 warmup steps. Training is done for 2 epochs with a batch size of 16.

We then finetune this model on Rotten Tomatoes (Pang & Lee, 2005), SST-2 (Socher et al., 2013), and Yelp (Zhang et al., 2015), and train with a learning rate of $5 \cdot 10^{-6}$ using a batch size of 16 and for 5, 5, and 2 epochs each. We subsample the data of Yelp by taking the first $20\%$ of the training data to ease computational burden.

We do not use any weight decay in pretraining or finetuning. The squared gradient approximation is calculated by doing one pass over the training data of each model and squaring the single-example gradients for ADAMW-SG.

For the batch solution, we finetune for 3 epochs on the concatenation of the above-mentioned training data after pretraining on IMDB as described above. Again, we use a learning rate of $5 \cdot 10^{-6}$ and a batch size of 16. Evaluation is done by averaging the accuracies over each individual dataset to weigh each dataset the same.

The simplex in Fig. 6 is obtained by sampling from a grid $\boldsymbol{\Delta}$ with spacing $0.05$. For the joint solution we use a spacing of $0.1$ due to the heavy computational load. The simple merged models are obtained using Eq. 2. For diagonal Gaussians, we use the Hessian-based weighting of Daheim et al. (2024).

### B.6 MERGING LLMS FOR MACHINE TRANSLATION

We finetune GEMMA-2B-it (Gemma Team, 2024b) on the IWSLT2017 de-en and fr-en splits (Cettolo et al., 2017). Due to the model size we use LoRA (Hu et al., 2022) to finetune the models which amounts to ca. 0.9M of new trainable parameters. The rest of the network is kept frozen. Accordingly, only the LoRA weights are merged and the base model untouched.

We train the models using IVON with a learning rate of 0.05, $(\beta_1, \beta_2) = (0.9, 0.99995)$, an effective sample size of $1 \cdot 10^7$ for the single-task and $2 \cdot 10^7$ for the multitask model. We clip gradients element-wise to $1 \cdot 10^3$ and to unit norm and use a weight decay of $10^{-6}$.

For the Hessian-weighted merging we use IVON-HESS. A comparison to using squared gradients instead is found in App. C.2.

For all experiments, we use a grid with equal spacing of $\alpha_1 \in [0.0, 0.05, \ldots, 1.0]$ and always set $\alpha_2 = 1.0 - \alpha_1$.

## C ADDITIONAL RESULTS

### C.1 MULTITASK LEARNING ON MNIST

We consider MNIST broken into three tasks, each consisting of a different and disjoint subset of classes. We use a logistic regression model in two settings: one imbalanced set where number of classes per tasks vary and a balanced set. Results are shown in Fig. 8 and Fig. 9, respectively. In both cases we compare isotropic Gaussian (Simple Merging), full Gaussian (Hessian-Weighted),

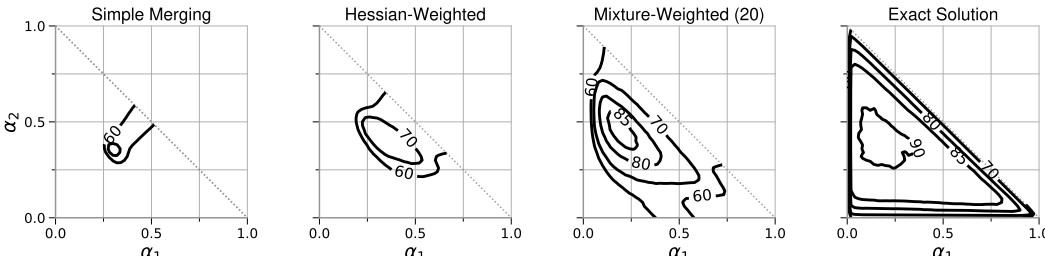

Figure 8: Multitask Learning on MNIST. As the posterior approximation gets more expressive the preview generated by the merged model resembles the exact solution better.

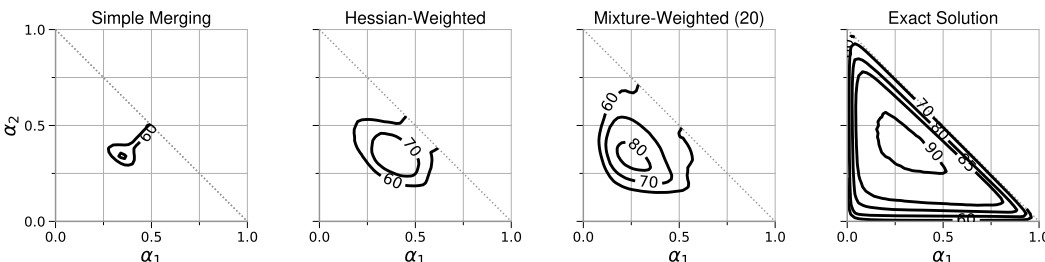

Figure 9: Similar to the previous setting, on balanced tasks improving the posterior approximation also will produce previews that capture more the the exact solution.

and mixture-of-Gaussian (Mixture-Weighted) posteriors. To compute the posterior approximations and surrogate functions, we use VON (Khan & Rue, 2023) and for mixture-of-Gaussians the joint learning algorithm from Lin et al. (2019), both with full Hessian. For Hessian-Weighted merging we use Eq. 10 for all $\alpha$, for the mixture-of-Gaussians we use the EM algorithm outlined in Eq. 12.

We find that better posterior approximations generally give better models but, more importantly, also more closely match the shape of the joint training solution. Simple Merging would only show very few weightings as good but in fact there is a large region with good weights which is better shown by better posterior approximations. Notably, the mixture-based approach following Eq. 12 also picks up the skew to the left shown in the joint solution for imbalanced tasks.

For the balanced setting there is no skew and the better combinations seem to concentrate around the center of the simplex, which we see in Figure Fig. 9 is captured by all methods, however the more complex posterior approximation allows Hessian-Weighting and Mixture-Weighting to show that multiple combinations even beyond the center are also interesting which Simple Merging fails to convey.

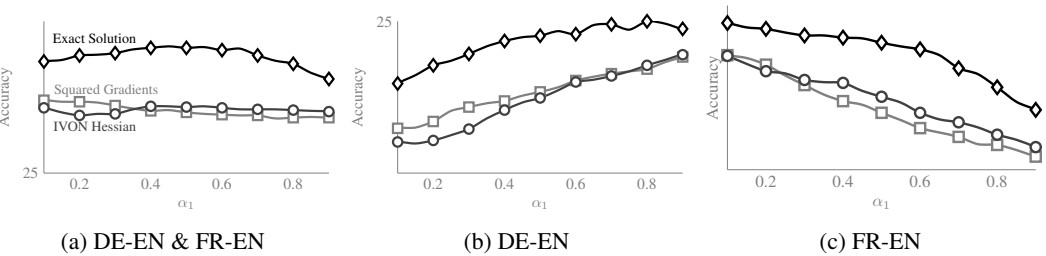

(a) DE-EN & FR-EN                (b) DE-EN                (c) FR-EN

Figure 10: Here, we merge LoRA-finetuned GEMMA-2B models trained on IWSLT2017de-en and IWSLT2017fr-en. We show a comparison of using the squared gradient approximation of the (diagonal) Fisher and the diagonal Hessian approximation obtained with IVON for Hessian-weighted merging. Both methods perform similarly and could be used effectively for previews.

| | Figure | Model | | Simple Merging | Hessian Weighted | Mixture Weighted | Multitask Finetuning |
|---|---|---|---|---|---|---|---|
| CV | Fig. 8 | Logistic | $\alpha_1$ | 0.34 | 0.42 | 0.14 | 0.12 |
| | | | $\alpha_2$ | 0.34 | 0.36 | 0.46 | 0.46 |
| | Fig. 9 | Logistic | $\alpha_1$ | 0.34 | 0.40 | 0.24 | 0.28 |
| | | | $\alpha_2$ | 0.34 | 0.34 | 0.34 | 0.42 |
| | Fig. 3 | ResNet-20 | $\alpha_1$ | 0.40 | 0.30 | 0.40 | 0.60 |
| | | | $\alpha_2$ | 0.20 | 0.40 | 0.20 | 0.10 |
| | Fig. 4 | ViT-B/32 | $\alpha_1$ | 0.35 | 0.35 | | 0.15 |
| | | | $\alpha_2$ | 0.35 | 0.35 | | 0.70 |
| | | ViT-B/32 | $\alpha_1$ | 0.35 | 0.35 | | 0.10 |
| | | | $\alpha_2$ | 0.40 | 0.45 | | 0.75 |
| NLP | Fig. 6 | RoBERTa | $\alpha_1$ | 0.10 | 0.15 | | 0.15 |
| | | | $\alpha_2$ | 0.55 | 0.50 | | 0.40 |
| | Fig. 7 | GEMMA-2B | $\alpha_1$ | 0.05 | 0.40 | | 0.45 |

Table 3: We report the best $\alpha$ combination found through previews for each experiment. As seen in Table 2, for most experiments, even if the $\alpha$ weights do not exactly match the multitask finetuned ones, accuracy is still high. Note that generating previews for several $\alpha$ using merging takes seconds and can guide the practitioner towards high accuracy weightings, while a full sweep through many combinations for multitask finetuning becomes prohibitive with larger models.

## C.2    COMPARISON OF HESSIAN APPROXIMATIONS FOR LLM MERGING

Fig. 10 shows a comparison of using the squared gradient approximation of the (diagonal) Fisher and the diagonal Hessian approximation obtained with IVON for Hessian-Weighted merging. Both methods are comparable and provide good previews for multitask finetuning. However, the Hessian approximation from IVON comes for free during training while the squared gradient approximation incurs overhead due to requiring an additional forward pass over at least a subset of the training data after training.

## C.3    WEIGHTS FOUND VIA PREVIEWS

Table 3 summarizes the $\alpha$ combinations that achieve the maximum accuracy according to each preview method. From the results in Table 2, we see that previews via merging obtain weights that achieve a comparably high accuracy, even if they do not always match the ones found by several runs of multitask finetuning. The accuracy also improves with better posterior approximations. Note that the cost to prepare the individual models per task to do previews is equivalent to one run of multitask finetuning. Afterwards, several weights can be tried without any extra training and at negligible overhead using merging. Exploring such weights through multitask finetuning becomes prohibitively expensive with larger models and larger amounts of tasks.

