# OpenReview forum: "How to Weight Multitask Finetuning? Fast Previews via Model Merging"
_ICLR.cc/2025/Conference — Submitted to ICLR 2025_

### Official Review · Reviewer_5Vwq · 2024-11-03

**Soundness:** 2
**Presentation:** 2
**Contribution:** 1
**Rating:** 3
**Confidence:** 4

**Summary:**

The paper focus on selecting optimal weights in multitask fine-tuning. The authors propose a Bayesian model-merging approach that previews different weighting configurations efficiently by leveraging model merging, thus reducing the need for exhaustive retraining on each configuration. The steps can be decomposed to:
1. Finetune $T$ models (denoted by $\boldsymbol\theta_t)$ each seprately over their own task $\ell_t(\boldsymbol{\theta}).$
2. Use Bayesian learning to build surrogate $\ell_t\approx\widehat{\ell}_t$ by using $\boldsymbol{\theta}_t.$
3. Create previews by fınetuning over $\sum_t\alpha_t\widetilde{\ell}_t$ for many $\boldsymbol{\alpha}$ values.

**Strengths:**

The Bayesian optimization based model-merging allows for quick performance previews by averaging parameters from separately trained task-specific models, avoiding full retraining.

**Weaknesses:**

- The method relies on a standard application of Bayesian optimization with a surrogate model, lacking novel contributions or distinctive techniques.

- In the steps listed between lines L148 and L151, while the three initial steps are understandable, the sequence is left open-ended. It’s unclear what action follows the third step—what process or computation does the algorithm proceed with after generating the previews?

- In the section between lines L270 and L280, the definitions and relationships among variables are unclear. For instance, the connection between $\hat{\pi}{tk}$ and $\pi{k}$ should be clarified. Additionally, if $h_{tk}$ is a scalar, it should not be in bold, as this can lead to confusion.

- There is a lack of a clear evaluation criterion. What mathematical definition is used to judge “better previews,” and what specific objective or performance metric is this algorithm designed to optimize?

**Questions:**

- The numbers in the figure (e.g., 90, 85, 80, ...) are labeled as accuracy, but it is unclear what these accuracy values represent. Are they calculated based on an equal weighting across three tasks? Please define the metrics and explain the weighting, if any, to clarify the interpretation.

- The three tasks listed—French, Spanish, and English—are somewhat unclear in meaning. What does “accuracy for English” specifically indicate in this context? Please provide a clear explanation of how accuracy is determined for each language task and what each result represents.

- As a reader, the takeaway from the experimental section remains ambiguous. There doesn’t appear to be a consistent or clearly stated conclusion. Could you provide a concise summary of the findings or implications drawn from this section to clarify its purpose and insights? Otherwise, the paper seems to give me an impression of an application of Bayesian optimization.

Please kindly try to answer my questions in both weakness and questions part. Thank you!

---

> ### Author Response · Authors · 2024-11-21
>
> We thank the reviewer for the questions, bellow we provide our answers.
>
> > **Q1:** The method relies on a standard application of Bayesian optimization with a surrogate model, lacking novel contributions or distinctive techniques
>
> **A1:** We strongly disagree. We do not use “Bayesian Optimization” which is a technique for global optimization and search and often uses Gaussian Process surrogates over function values (no gradients involved). In our work, we use exponential-family posterior approximations as surrogates, which is fundamentally different from the Gaussian-Process surrogates used in Bayesian Optimization.
>
> > **Q2:** In the steps listed between lines L148 and L151, while the three initial steps are understandable, the sequence is left open-ended. It’s unclear what action follows the third step—what process or computation does the algorithm proceed with after generating the previews?
>
> **A2:** The main use of the preview is to choose the good $\boldsymbol\alpha$ values and use them to run full multitask finetuning. We have added a few lines to clarify this in the revised version; see L39, L91, and L145. The preview is also useful to analyze the trade-offs among tasks.
>
> >**Q3:** In the section between lines L270 and L280, the definitions and relationships among variables are unclear. For instance, the connection between $\hat{\pi}\_{tk}$  and $\pi\_{k}$ should be clarified. Additionally, if $\mathbf{h}\_{tk}$ is a scalar, it should not be in bold, as this can lead to confusion.
>
> **A3:** Please see the line right after Eq. 12, where the definition of $\hat{\pi}\_{tk}$ is given. Also, $\mathbf{h}\_{tk}$ is the diagonal of the Hessian, and not a scalar.
>
>
> >**Q4:** There is a lack of a clear evaluation criterion. What mathematical definition is used to judge “better previews,” and what specific objective or performance metric is this algorithm designed to optimize?
>
> **A4:** See Table 1 where we use the MSE between the Exact-Solution contour and the previews obtained using model merging; this is also mentioned in Line 360. This is a clear objective criteria (although other metrics can be used). As the objective of the algorithm is concerned, the proposed algorithm tries to find the best merged model for each $\boldsymbol\alpha$ separately. This is done by using a variational approach (that is, by minimizing KL).
>
> > **Q5:** The numbers in the figure (e.g., 90, 85, 80, ...) are labeled as accuracy, but it is unclear what these accuracy values represent. Are they calculated based on an equal weighting across three tasks? Please define the metrics and explain the weighting, if any, to clarify the interpretation.
>
> **A5:** Yes, we report test accuracy over the test set containing all the tasks together.
>
> >**Q6:** The three tasks listed—French, Spanish, and English—are somewhat unclear in meaning. What does “accuracy for English” specifically indicate in this context? Please provide a clear explanation of how accuracy is determined for each language task and what each result represents.
>
> **A6:** We are not sure which figure/result the reviewer is referring to; there is no “French, Spanish and English” together in any figure or experiment. For example, Fig. 1 refers to “Spanish, German, French” (no English). Similarly, Fig. 8 contains “German, English, French” (no Spanish). In general, “accuracy” in a language (like English) represents the model accuracy evaluated by the BLEU metric, which is a standard text-generation evaluation metric that measures overlap statistics between a generated and ground-truth output. We could clarify further if you tell us the exact section where you find the problem.
>
> > **Q7:** As a reader, the takeaway from the experimental section remains ambiguous. There doesn’t appear to be a consistent or clearly stated conclusion. Could you provide a concise summary of the findings or implications drawn from this section to clarify its purpose and insights? Otherwise, the paper seems to give me an impression of an application of Bayesian optimization.
>
> **A7:** The summary is given in Table 1 and the summary of the finding is stated in L361-362. Every section also contains such a summary (for example, L374 in Sec 4.1).

---

> > ### Author Response · Authors · 2024-11-25
> > **Gentle Reminder.**
> >
> > Dear Reviewer, since the discussion period will be closing soon, please let us know if there are any further questions.

---

> ### Comment · Reviewer_5Vwq · 2024-12-03
> **I decide to maintain the origin score**
>
> Dear Authors,
>
> Thank you for sharing your work. I have carefully reviewed the manuscript and would like to provide feedback for consideration.
>
> ### Feedback:
>
> 1. (Q1) Clarification on Bayesian Optimization:
>    - I understand you are using "exponential-family posterior approximations as surrogates." However, this still an application of Bayesian Optimizations.
>
> 2. (Q2) Evaluation Methodology Across Tasks ($T$):
>    - It is stated that "the main use of the preview is to choose the good alpha values and use them to run full multitask finetuning." However, I am unclear about the evaluation methodology across tasks.
>    - Specifically, is the evaluation metric $\sum_t l_{\text{test},t}$, $\sum_t \alpha_t l_{\text{test},t}$, or something else entirely? This point requires further elaboration to improve clarity.
>
> 3. (Q4) Model Merging Comparisons:
>    - You propose a new model merging method, yet there is no direct performance comparison with existing methods (e.g. ties-merging, DELLA-Merging, Ada-Merging, DARE). This omission weakens the practical relevance of the study.
>    - While MSE between the Exact-Solution contour and the previews is a useful intermediate metric, performance ultimately matters more. Including metrics like win rate or average performance scores for merged models would significantly strengthen the narrative.
>
> 4. (Q7) Posterior Approximation and Multitask Finetuning:
>    - Regarding the statement:
>      > "Overall, better posterior approximations match multitask finetuning more closely both in terms of overall shape of the solution (Table 1) and the performance of the best weightings (Table 2)."
>      - Why does this matter in practical terms? If your proposed method consistently outperforms other merging techniques, that would explain why the Bayesian Optimization framework used. However, without direct comparisons to baseline merging methods, the importance of this step is less compelling.
>
> ### Additional Comments
> 1. Experimental Design:
>    - L396: You describe the experiment using ResNet-20 on CIFAR-10 with three tasks constructed from different sets of classes. It would be helpful to explicitly define the criteria for selecting these tasks to avoid concerns about cherry-picking.
>    - Since the computational cost of this experiment appears low, I suggest iterating over all task combinations and reporting averages and standard deviations to improve the robustness and believability of results.
>    - The study focuses on a low-dim task space. How does the method scale to higher-dim search spaces, and what is the search efficiency? (Existing baselines are on 8 number of tasks rather than 2-3.)
>    - The absence of direct comparisons to existing baseline methods on the performance of merged models is a significant gap. Such comparisons are crucial for validating your approach.
>    - L177: Provide more detail on why the Hessian is regarded as an identity matrix in this context.
>
> 2. Mathematical Notation Casualty:
>    - While the mathematics is well-developed, the notation can feel casual and at times ambiguous, making it difficult to follow.
>      - Example 1: $\theta_{\alpha}$ appears at L105 but is not defined. Readers are left to infer its meaning. Additionally, subscripts of $\theta$ alternate between $t$ and $\alpha$, which are not equivalently positioned conceptually.
>      - Example 2: $\widehat{\mathcal{R}}_0$ is introduced at L161 without context. Why is there a hat, and why is there a 0? There is no $\mathcal{R}_0$ in the whole paper. I can guess it can be regarded as the unnormalized probability at step 0. But it could be hard to follow if readers need to guess everywhere. Clear definitions are necessary to avoid unnecessary guesswork. (This also applies to the main text. E.g. L180, when you say one point, what does it mean? After reading a few paragraphs, I am able to guess that the "point" here means model weights, but I think it jams the reading.)
>      - Example 3: The use of $\Rightarrow$ in equations (8), (9), (11), and L257 lacks formal consistency. I understand that when doing induction on draft papers, it is very common to use $\Rightarrow$. However, I think this is not very formal in the manuscript to be published. Furthermore, the numbering of equations (L257) is incomplete.
>
> 3. Experimentation Completeness:
>    - While the authors have conducted extensive experiments, the lack of baseline comparisons is a significant shortfall for an application-focused paper. Incorporating baselines is essential to demonstrate the practical utility of the proposed method.
>
> ### Overall Assessment:
> This paper demonstrates an application of modified Bayesian Optimization for model merging. However, because of the lack of comparisons with baseline methods, incomplete clarity in certain areas, and casual mathematical notation reduce its readiness, I decide to maintain the origin score.
>
> Thank you for your understanding.
>
> P.S.
> - Typo
>    - L187: "but how can be design them?"

---

> > ### Author Response · Authors · 2024-12-04
> > **Response part 1**
> >
> > The reviewer is dismissive of our work on two main points which we disagree with.
> >
> > - First, that our work is an “application of Bayesian optimization” which is incorrect: our work uses variational (Bayesian) learning to build surrogates for model merging which is novel and not simply an application.
> > - Second, the reviewer claims our experiments are weak because comparisons to baselines are missing (“cherry-picked”, “improve robustness and believability”, “[is] significant gap”). We disagree strongly and respond to those below.
> >
> > We must also point out that the comments are posted just 1 day before the deadline, and asks for more experiments which is against the recommendation of ICLR guidelines. Note that the reviewer originally did not ask for any such experiments in the review. We also do not agree with the reviewer that such experiments are necessary.
> >
> > Below, we provide a detailed response.
> >
> > **1. “Still an application of Bayesian optimization”**
> >
> > > “I understand you are using ‘exponential-family posterior approximations as surrogates.’ However, this still an application of Bayesian Optimizations.”
> >
> > > “If your proposed method consistently outperforms other merging techniques, that would explain why the Bayesian Optimization framework used.”
> >
> > We respectfully disagree that our method is “an application of Bayesian Optimizations”. The reviewer is incorrect: [Bayesian Optimization](https://en.wikipedia.org/wiki/Bayesian_optimization) refers to a specific global optimization method, which we do not use. For instance, see Eq. 1 in [1] which is a (popular) review paper on this topic from 2015. Our approach is not simply an application of this approach. Ours is a novel approach in the context of model merging: there is no existing approach that uses variational learning to build surrogates for model merging. Note that these surrogates are different from surrogate models used in Bayesian Optimization, both in their form and the way we use them.
> >
> > [1] Shahriari et al. Taking the Human Out of the Loop: A Review of Bayesian Optimization. (2015) https://ieeexplore.ieee.org/document/7352306
> >
> > **2. A Question about evaluation metric:**
> >
> > > is the evaluation metric $\sum_t l_{\text{test},t}$, $\sum_t \alpha_t l_{\text{test},t}$, or something else entirely?
> >
> > The evaluation metric is $\sum_t l_{\text{test},t}$. The alpha weights are not used at test time. The goal is to find alpha that leads to a good performance over all tasks.
> >
> > **3. Lack of comparisons to existing model-merging baselines**
> >
> >  > “You propose a new model merging method, yet there is no direct performance comparison with existing methods (e.g., ties-merging, emr-merging, Ada-Merging). This omission weakens the practical relevance of the study.”
> >
> > >While MSE between the Exact-Solution contour and the previews is a useful intermediate metric, performance ultimately matters more. Including metrics like win rate or average performance scores for merged models would significantly strengthen the narrative.
> >
> > > “Regarding the statement: "Overall, better posterior approximations match multitask finetuning more closely both in terms of overall shape of the solution (Table 1) and the performance of the best weightings (Table 2)." Why does this matter in practical terms? [...] Without direct comparisons to baseline merging methods, the importance of this step is less compelling.”
> >
> > > “The absence of direct comparisons to existing baseline methods on the performance of merged models is a significant gap. Such comparisons are crucial for validating your approach.”
> >
> > > “While the authors have conducted extensive experiments, the lack of baseline comparisons is a significant shortfall for an application-focused paper.”
> >
> > The reviewer did not ask for such experiments in the original review and now they are asking this only 1 day before the deadline. We also disagree that such experiments are necessary. The reviewer misunderstands the goal of the paper which is to find good alpha values, not to propose a new merging method. The focus is not the quality of a particular merging method at a specific alpha value but its ability to estimate the true shape well over a range of them, which is a practical goal.
> >
> > We do not claim that other merging methods are worse and ours are better. We simply show that better previews are obtained with better merging methods that use a more flexible updating scheme (corresponding to a flexible posterior approximation). A comparison with other merging methods can be done, but it would not support/reject our findings. Therefore, it is unnecessary.

---

> > > ### Author Response · Authors · 2024-12-04
> > > **Response part 2**
> > >
> > > **4. Suggestions for further experimentations**
> > >
> > > > “You describe the experiment using ResNet-20 on CIFAR-10 with three tasks constructed from different sets of classes. It would be helpful to explicitly define the criteria for selecting these tasks to avoid concerns about cherry-picking. [...] Since the computational cost of this experiment appears low, I suggest iterating over all task combinations and reporting averages and standard deviations to improve the robustness and believability of results.”
> > >
> > > We do not cherry-pick tasks. We show results over a wide range of settings where trends are stable and consistent. The results are robust (and believable). Asking for further experimentation is unfair 1 day before the deadline. The cost of re-running the ResNet-20 experiments on all possible task combinations is actually high and not feasible so close to the deadline. This goes against the recommendation of ICLR guidelines.
> > >
> > > > “The study focuses on a low-dim task space. How does the method scale to higher-dim search spaces, and what is the search efficiency? (Existing baselines are on 8 number of tasks rather than 2-3.)”
> > >
> > > Regarding large number of tasks, we have not explored this here but see our response to Reviewer 9n4H where we mention future work to be done in this direction.
> > >
> > > **5. Concerns about notation and a minor question**
> > >
> > > 1. “$\theta_{\alpha}$ appears at L105 but is not defined.”
> > > 2. “ $\widehat{\mathcal{R}}_0$ is introduced at L161 without context. [...] There is no $\mathcal{R}_0$ in the whole paper.”
> > > 3. “E.g. L180, when you say one point, what does it mean? After reading a few paragraphs, I am able to guess that the "point" here means model weights, but I think it jams the reading.”
> > > 4. “The use of $\Rightarrow$ in equations (8), (9), (11), and L257 lacks formal consistency. I understand that when doing induction on draft papers, it is very common to use $\Rightarrow$. However, I think this is not very formal in the manuscript to be published. Furthermore, the numbering of equations (L257) is incomplete.”
> > >
> > > All these points refer to small typos that are easy to fix. They do not imply that our notation is casual or that we were careless about it.
> > >
> > > 1. This is incorrect. $\theta_{\alpha}$ is defined in L105.
> > > 2. There is a small typo in L163. Note that $\mathcal{R}_0$ does appear later in L208.
> > > 3. One point in L180 means $\theta_t$.
> > > 4. We disagree about $\Rightarrow$ and equation numbering. These are subjective preferences.
> > >
> > >
> > > > “L177: Provide more detail on why the Hessian is regarded as an identity matrix in this context.”
> > >
> > > The Hessian is *not* regarded as the identity matrix. Rather, Task Arithmetic corresponds to a simplistic Taylor’s approximation.

---

### Official Review · Reviewer_t1Ce · 2024-11-03

**Soundness:** 3
**Presentation:** 2
**Contribution:** 3
**Rating:** 6
**Confidence:** 2

**Summary:**

The paper leverages model merging to create fast previews for identifying optimal weights for each task in multitask finetuning. This approach enhances efficiency and effectiveness by providing quick guidance on reasonable search areas. Specifically, the paper proposes a Bayesian method that employs more flexible posteriors to improve the quality of these previews. Various experiments demonstrate that the method can effectively preview suitable multitask finetuning weightings.

**Strengths:**

The method can alleviate the costly burden of exhaustive searching in multitask finetuning by reusing the parameters of models trained on each task separately, eliminating the need for retraining to find the optimal weights. It also provides insights that focusing solely on the best-performing weights in model merging may not always yield high-quality previews.

**Weaknesses:**

It is unclear whether the weights chosen for each task by the method will be used for further multitask finetuning. If they are, the method appears to be less computationally efficient because it requires training the models for each task initially. If they are not, the accuracy of the merged model seems to lag significantly behind joint training on average, as shown in Figures 4, 5, and 6, suggesting that a simple grid search of joint training could outperform model merging. Additionally, the paper lacks comparisons with other reweighting or model merging methods.

**Questions:**

On average, how long does it take to converge for Simple Merging, Hessian-Weighted Merging, and Mixture-Weighted Merging?

---

> ### Author Response · Authors · 2024-11-21
>
> We thank the reviewer for the questions, bellow we provide our answers.
>
> >**Q1:** It is unclear whether the weights chosen for each task by the method will be used for further multitask finetuning.
>
> **A1:** Yes, the main idea is to choose weights using previews and then use them to jointly perform multitask finetuning only once. We have added a few lines in Sec 1 and 3 to clarify this; see L39, L91, and L145 in the revised version.
>
> >**Q2:** If they are, the method appears to be less computationally efficient because it requires training the models for each task initially.
>
> **A2:** We believe there is a misunderstanding here. Our approach aims to reduce the cost of exhaustive search over the alpha-space by using separately trained models on each task. Separate training needs to be done just once and is sufficient to quickly generate the estimates for multitask finetuning with a variety of alpha-values. For example, for the experiment in Fig. 5, it takes seconds to create estimates compared to around 50 mins to run a single multitask finetuning run. Doing such finetuning on 100s of tasks is too expensive. In our approach, training each single-task model takes around 15 mins, and this only needs to be done. With this, creating estimates just takes seconds. Overall, a single run of multitask finetuning via previews has similar runtime to two runs of multitask finetunings (for instance, two settings of $\boldsymbol\alpha$ values)
>
> We have included such exact runtimes in L431 and L523.
>
> >**Q3:** If [the method is not used for multitask finetuning, then] the accuracy of the merged model seems to lag significantly behind joint training on average, as shown in Figures 4, 5, and 6, suggesting that a simple grid search of joint training could outperform model merging. Additionally, the paper lacks comparisons with other reweighting or model merging methods.
>
> **A3:** This question is not valid because our goal is to just select good weights and then run multitask finetuning with those weights. See our response to Q1.
>
> >**Q4:** On average, how long does it take to converge for Simple Merging, Hessian-Weighted Merging, and Mixture-Weighted Merging?
>
> **A4:** For both simple and Hessian-weighted merging, we have closed form expressions so there is no notion of convergence. It takes seconds to do this operation; see L431 and L523 for two examples. For mixture-weighted merging, we need to do Hessian-weighted merging a few times, but often very few, for example, the ViT merging experiments required 2 iterations.

---

> > ### Comment · Reviewer_t1Ce · 2024-11-24
> > **Official Comment by Reviewer t1Ce**
> >
> > Thank you for your response, But I still have some questions, such as the description "Overall, a single run of multitask finetuning via previews has similar runtime to two runs of multitask finetunings" Is it only applicable when there are fewer tasks? For example, if it takes about 15 minutes to train a single-task model, then training for 100 tasks (a more extreme case) would require a total of 15*100=1500 minutes. This is equivalent to the cost of 30 multi-task training runs if it takes 50 minutes for a single multi-task finetuning run. Additionally, if the training time of each task is longer, the computational cost of training the model for each task at first increases significantly.
> >
> > Therefore, I wonder whether a few grid searches are enough to obtain a good set of weights for multi-task fine-tuning. For example, in the scenario above, using your approach would require training a model for each of the 100 tasks and then determining the final multi-task fine-tuning weights. In contrast, a grid search might only need to be performed 10 times, resulting in a computational cost of just 50*10=500 minutes.
> >
> > Looking forward to your response!

---

> > > ### Author Response · Authors · 2024-11-25
> > >
> > > Thanks for your questions. We do not think that the usefulness and impact of our work is diminished for a large number of tasks:
> > > 1. First, we do not agree with your calculations regarding the costs. If the number of tasks grows, the time for multi-task finetuning *also* grows. For instance, in your example with 100 tasks, one single run of multi-task finetuning will also take roughly 1500 mins (not just 50 mins as you suggested). This is because the size of the dataset will be much larger.
> > > 2. Second, we stress that the cost of *all* individual single-task finetuning is similar to the cost of *one* multi-task finetuning for all our experiments, which is affordable. We do not expect this cost to increase significantly (e.g. exponentially) as the number of tasks grows.
> > > 3. In contrast, the cost of a grid search increases exponentially as the number of tasks grows. For example, for 5 grid points, 100 tasks would require $5^{100}$ (not 10 times as suggested because picking 10 values out of $5^{100}$ is challenging). In our approach, we do not need such a grid search: we simply train 100 single-task models whose cost would be fairly similar to one run of multitask training. Computing previews over all $\boldsymbol\alpha$ values will take very little time in comparison.
> > > 4. Ideally, for large tasks we want to do a gradient based optimization over the preview. This is possible with the new model-merging techniques proposed in this paper. We hope to explore this in the future.
> > >
> > > We are happy to answer any more questions you might have. Thanks!

---

> > > > ### Comment · Reviewer_t1Ce · 2024-11-25
> > > > **Official Comment by Reviewer t1Ce**
> > > >
> > > > Thank you for your detailed response, which addresses most of my concerns. I will raise the score to 6.

---

> > > > > ### Author Response · Authors · 2024-11-28
> > > > > **Thank you!**
> > > > >
> > > > > We thank the reviewer for raising their score.

---

### Official Review · Reviewer_9n4H · 2024-11-04

**Soundness:** 3
**Presentation:** 3
**Contribution:** 2
**Rating:** 6
**Confidence:** 4

**Summary:**

This paper presents a novel approach for efficiently identifying task weightings in multitask finetuning by quickly previewing/estimating the performance of different tasks using Bayesian model merging techniques. The authors propose to use Bayesian posteriors as surrogates to estimate performance across various weightings. In addition, the exponential family distributions are further introduced to create more flexible and accurate surrogates. The approach is validated through experiments on diverse tasks including a series of CV and NLP tasks to demonstrates that the effectiveness of the previews on multitask weightings.

**Strengths:**

1. The paper is overall well-written and the motivation is clear.
2. The Bayesian learning method establish a rigorous mathematical framework, providing a principled approach to derive and enhance the weighting selection strategy on model merging.

**Weaknesses:**

- There is limited quantitative analysis of the computational savings(or practical time savings) across different tasks compared to exhaustive grid search. The mixture-based approach may become computationally intensive for large models or when dealing with a large number of tasks.
- Some trends have not been precisely captured by either Hessian-weighted merging or simple merging (Fig 6(b) & Fig 6(c)).

**Questions:**

Questions have been covered in the weaknesses section.

---

> ### Author Response · Authors · 2024-11-21
>
> We thank the reviewer for the questions, bellow we provide our answers.
>
> >**Q1:** There is limited quantitative analysis of the computational savings(or practical time savings) across different tasks compared to exhaustive grid search.
>
> **A1:** We have added a few lines in Sec 3.5 (see L314-321 of the revised version) for computational overheads. We have also added absolute runtime in Sec 4.3 and 4.5 (L431 and L524). As an example, for experiments in Fig.5, joint multitask finetuning takes around 50 minutes for each $\alpha$ value, but merging takes only seconds (plus around 14 to 17 minutes for finetuning on each task separately). Overall, a single run of multitask finetuning with previews has similar runtime to two runs of multitask finetunings (around 100 minutes for Fig. 5).
>
> >**Q2:** The mixture-based approach may become computationally intensive for large models or when dealing with a large number of tasks.
>
> **A2:** We agree that the computation increases a lot but the approach still remains much cheaper than exhaustive search over the $\boldsymbol\alpha$-space by explicitly training on all tasks together. For instance, for Fig. 5, merging with mixtures still takes a few seconds, while the joint multitask training takes around 50 minutes.
>
> >**Q3:** Some trends have not been precisely captured by either Hessian-weighted merging or simple merging (Fig 6(b) & Fig 6(c)).
>
> **A3:** In Fig. 6a and 6b, we see clear improvements by Hessian-weighted over simple averaging for $\alpha$ values from 0 to 0.5. We agree that in 6c the trends are similar. In general, it is true that we may not always see gains in accuracy of the preview when increasing the posterior class.

---

### Official Review · Reviewer_rd2S · 2024-11-04

**Soundness:** 3
**Presentation:** 3
**Contribution:** 3
**Rating:** 8
**Confidence:** 3

**Summary:**

The paper proposes to create previews to identify optimal weightings for tasks in multi-task finetuning setting. The approach creates previews of performance under various weight configurations and uses bayesian model-merging approach to combine separately fine-tuned models via baysian posteriors model weights merging. The proposed method allows practitioners to get an rough performance estimation without trying various weighting exhaustively.  The approach is experimented on both vision and language tasks, demonstrating its efficacy across different model types and tasks.

**Strengths:**

1. Motivation of this paper is strong. Finding proper balance between tasks in multitask finetuning has always been challenging for heavy training jobs (large models or large amount of data). The paper focuses on a very important research problem.
2. The proposed approach is novel - a bayesian merging strategy using flexible posteriors (gaussian) and use misxtured-based merging for better accuracy.
3. The paper is well written.
4. The proposed approach is effective on various tasks and architectures, demonstrating good generalization.

**Weaknesses:**

1. There isn't enough analysis of computation overhead of the proposed approach.
2. The paper doesn't have enough discussions of its limitations. Presenting limitations would give readers an better and more comprehensive understanding of your approach.

**Questions:**

Can this approach be widely adopted for multitask learning training?


Do you plan to release the code? Is it easy for people in the community to use your code for their own multitasks training jobs?

What are the limitations?

---

> ### Author Response · Authors · 2024-11-21
>
> We thank the reviewer for the positive score. Below we provide answers to their questions.
>
> >**Q1:** There isn't enough analysis of computation overhead of the proposed approach.
>
> **A1:** We have added a few lines in Sec 3.5 (see L314-321 of the revised version) for computational overheads. We have also added absolute runtime in Sec 4.3 and 4.5 (L431 and L524). To summarize, the IVON-Hess method is the cheapest, there is virtually no cost to compute the parameters for each finetuned model and their associated diagonal Hessian estimate (denoted by $h_t$). Merging requires only element-wise vector multiplication to get $h_t \cdot \theta_t$ and elemen-twise addition to add them all together. All such computations are very cheap and scale linearly in the number of parameters.
>
> The multiIVON-Hess, requires multiple (say up to $K$) IVON-Hess runs to obtain $K$ components and multiple iterations (say $I$), which increases the cost by $K\cdot I$ times. The AdamW-SG method requires an additional pass through the data to get the “squared-gradient” estimate of the Hessian.
>
> We will add these details in the camera ready version if accepted.
>
> >**Q2:** Can this approach be widely adopted for multitask learning training?
>
> **A2:** We believe it can. Choosing good weights through exhaustive search is time-consuming and previews are excellent cheap alternatives. Being able to choose better weights directly impacts accuracy. For example, in Fig. 4, the best accuracy of joint training (Exact Solution) is around 70% which goes down quickly to 65% and lower as we move away from the best weights.
>
> >**Q3:** Do you plan to release the code? Is it easy for people in the community to use your code for their own multitask training jobs?
>
> **A3:** Yes, we will release the code upon acceptance. We also plan to integrate the code on the mergekit library (https://github.com/arcee-ai/mergekit/).
>
> >**Q4:** What are the limitations? The paper doesn't have enough discussions of its limitations. Presenting limitations would give readers a better and more comprehensive understanding of your approach.
>
> **A4:** We have included a limitations statement as Section 5 (Line 538 in the new version). The main limitation is that the number of components and training steps for the mixture surrogates may be large in some cases. In many cases, good results are obtained with 5-10 steps but this is not always the case. This could also be prohibitive when storing more than one copy of the model is too costly.

---

### Author Response · Authors · 2024-11-28
**Improved manuscript**

We thank all the reviewers for the discussion. We found the feedback valuable and have made some more modifications to the PDF.
Here is a summary of changes.
1. We improved the writing of Sec 3.4 (especially Eq. 8 ) and also improved the appendix.
2. We improved the description in Sec 3.5 clarifying computations of each method.
3. We added Table 2 summarizing the maximum accuracies of all experiments and comparing them to those of the Multitask finetuning.

---

### Meta-Review · Area_Chair_b46p · 2024-12-21

**Metareview:**

The paper has received a mixed reviews: 8, 3, 6, 6. In this paper, the authors proposes a Bayesian approach to enhance multitask finetuning by using model merging techniques to provide fast previews of weighting strategies for tasks. The proposed method introduces Bayesian surrogates and leverages exponential-family distributions to create previews that guide task weight selection. The paper provides experimental results across various benchmarks, including computer vision, natural language processing, and machine translation tasks, suggesting that the proposed method can approximate multitask finetuning outcomes with reduced computational costs.

Strengths
- The idea of employing Bayesian model merging to provide guidance for multitask finetuning is an interesting idea of using Bayesian principles.
- The proposed mechanism offers potential computational savings, especially for large-scale multitask learning problems.
- The proposed method seems can be applied across diverse domains, showcasing its generalizability.

Area for improvement:
- Theoretical novelty seems missing: reviewers noted that the methodological contributions rely heavily on existing techniques (e.g., Bayesian posterior merging, exponential-family distributions) without sufficient theoretical innovation.
- While experiments span multiple benchmarks, reviewers found them insufficiently rigorous. The results often do not decisively support the claims, either on performance improvement or computational efficiency.
- The comparison to existing multitask learning methods should also be more comprehensive. Reviewers pointed out that the practical impact of the proposed approach is not clear.
- Clarity and presentation: reviewers mentioned that the paper can be better organized/structured and better presented. The current version makes it challenging to discern the primary contributions and how they compare to existing methods.
- The proposed method in real-world scenarios remains unclear. The computational savings, while promising, may not outweigh the limited performance gains observed in the experiments.

While the idea of using Bayesian model merging for multitask finetuning is intriguing, the novelty, rigor, or impact of paper should be further improved to justify acceptance. The authors should strengthen the experimental evidence and enhance the theoretical innovation to better demonstrate the paper's potential contribution to the field and consider resubmitting.

**Additional Comments On Reviewer Discussion:**

Thanks to the authors and reviewers, there has been active and constructive discussion for this paper. After the discussion, several key issues remain unresolved, such as the concerns raised by Reviewer 5Vwq. These include the lack of sufficient theoretical novelty, the limited robustness and rigor of the experimental evidence, and the need for more comprehensive comparisons with state-of-the-art multitask finetuning approaches. While the rebuttal addressed some minor points and provided clarifications, the core challenges highlighted by reviewers seem to persist, leaving room for further improvement before resubmission.

---

### Decision · Program_Chairs · 2025-01-22

Reject